# BATCH SIZE-INVARIANCE FOR POLICY OPTIMIZATION

## ABSTRACT

We say an algorithm is *batch size-invariant* if changes to the batch size can largely be compensated for by changes to other hyperparameters. Stochastic gradient descent is well-known to have this property at small batch sizes, via the learning rate. However, some policy optimization algorithms (such as PPO) do not have this property, because of how they control the size of policy updates. In this work we show how to make these algorithms batch size-invariant. Our key insight is to decouple the proximal policy (used for controlling policy updates) from the behavior policy (used for off-policy corrections). Our experiments help explain why these algorithms work, and additionally show how they can make more efficient use of stale data.

## 1 INTRODUCTION

Policy gradient-based methods for reinforcement learning have enjoyed great success in recent years. The stability and reliability of these methods is typically improved by controlling the size of policy updates, using either a "trust region" (TRPO) or a surrogate objective (PPO) (Schulman et al., 2015; 2017). The usual justification for this is that we cannot trust updates that take us too far from the policy used to collect experience, called the *behavior policy*. In this work we identify a subtle flaw with this: the behavior policy is irrelevant to the justification. Instead, what matters is that we control *how fast* the policy is updated, or put another way, that we approximate the natural policy gradient (Kakade, 2001).

Our key insight is that the "old" policy in these methods serves two independent purposes. The first purpose is for off-policy corrections, via importance sampling, for which the old policy must be the behavior policy. The second purpose is to control the size of policy updates, for which the old policy can be any recent policy, which we call the *proximal policy*. It does not matter whether the proximal policy is also the behavior policy; it only matters *how old* the proximal policy is. We demonstrate this by running PPO with stale data collected using a policy from multiple iterations ago, which causes performance to quickly degrade unless the proximal policy is decoupled from the behavior policy.

Our insight allows us to make PPO *batch size-invariant*, meaning that when the batch size is changed, we can preserve behavior, as a function of the number of examples processed, by changing other hyperparameters (as long as the batch size is sufficiently small). We achieve this by using an exponentially-weighted moving average (EWMA) of the policy network's weights as the network for the proximal policy. Batch size-invariance has been studied many times before (see Section 3.1), sometimes under the name "perfect scaling". It is of practical benefit when we wish to increase the batch size to reduce gradient variance, but computational resources such as GPU memory do not allow this. In such a situation, we can instead adjust other hyperparameters formulaically, thereby spreading out the increased computational load over time.

The remainder of the paper is structured as follows.

- In Section 2, we explain the difference between the proximal and behavior policies, and show how to decouple them in PPO's objectives.
- In Section 3, we explain the concept of batch size-invariance, and how it applies to SGD and Adam (Kingma & Ba, 2014).
- In Section 4, we introduce PPO-EWMA and PPG-EWMA, variants of PPO and PPG (Cobbe et al., 2020) that make use of our decoupled objectives, and show how to make them batch size-invariant at small batch sizes.

- In Section 5, we provide experimental evidence for our central claims: that decoupling the proximal policy from the behavior policy can be beneficial, and that it allows us to achieve batch size-invariant policy optimization.
- Finally, in Section 6, we discuss the theoretical and practical implications of our results.

## 2 DECOUPLED POLICY OBJECTIVES

In this section we explain the difference between the proximal and behavior policies, and introduce new versions of PPO's objectives in which they have been decoupled.

PPO alternates between sampling data through interaction with the environment, and optimizing a surrogate objective. The policy used for sampling is denoted $\pi_{\theta_{\text{old}}}$, and is used by the objective in two different ways. This is easiest to see with the KL penalized objective (Schulman et al., 2017, equation (8)):

$$L^{\text{KLPEN}}(\theta) := \hat{\mathbb{E}}_t \left[ \frac{\pi_\theta(a_t \mid s_t)}{\pi_{\theta_{\text{old}}}(a_t \mid s_t)} \hat{A}_t - \beta \, \text{KL} \left[ \pi_{\theta_{\text{old}}}(\cdot \mid s_t), \pi_\theta(\cdot \mid s_t) \right] \right],$$

where $\hat{A}_t$ is an estimator of the advantage at timestep $t$, and $\hat{\mathbb{E}}_t [\ldots]$ indicates the empirical average over a finite batch of timesteps $t$. The first use of $\pi_{\theta_{\text{old}}}$ in this expression is as part of an importance sampling ratio. In order for the policy gradient estimate to be unbiased, this policy needs to be the one that was used for sampling, so we call this the *behavior policy* $\pi_{\theta_{\text{behav}}}$. The second use of $\pi_{\theta_{\text{old}}}$ is as a recent target to pull the current policy towards, so we call this the *proximal policy* $\pi_{\theta_{\text{prox}}}$.

Our key insight is that **the proximal policy need not equal the behavior policy**. As we will show experimentally, it matters *how old* the proximal policy is, but it does not matter whether or not the proximal policy was used for sampling.

We therefore define the *decoupled KL penalized objective*

$$L^{\text{KLPEN}}_{\text{decoupled}}(\theta) := \hat{\mathbb{E}}_t \left[ \frac{\pi_\theta(a_t \mid s_t)}{\pi_{\theta_{\text{behav}}}(a_t \mid s_t)} \hat{A}_t - \beta \, \text{KL} \left[ \pi_{\theta_{\text{prox}}}(\cdot \mid s_t), \pi_\theta(\cdot \mid s_t) \right] \right],$$

where $\pi_{\theta_{\text{behav}}}$ is the policy used for sampling, and $\pi_{\theta_{\text{prox}}}$ is a recent policy yet to be specified.

It is less obvious how to decouple the clipped PPO objective, because $\pi_{\theta_{\text{old}}}$ only appears once in that expression (Schulman et al., 2017, equation (7)):

$$L^{\text{CLIP}}(\theta) := \hat{\mathbb{E}}_t \left[ \min \left( r_t(\theta) \hat{A}_t, \text{clip}(r_t(\theta), 1 - \epsilon, 1 + \epsilon) \hat{A}_t \right) \right],$$

where $r_t(\theta) := \frac{\pi_\theta(a_t|s_t)}{\pi_{\theta_{\text{old}}}(a_t|s_t)}$. However, we can rewrite this objective as

$$L^{\text{CLIP}}(\theta) = \hat{\mathbb{E}}_t \left[ \frac{1}{\pi_{\theta_{\text{old}}}} \min \left( \pi_\theta \hat{A}_t, \text{clip}(\pi_\theta, (1 - \epsilon) \pi_{\theta_{\text{old}}}, (1 + \epsilon) \pi_{\theta_{\text{old}}}) \hat{A}_t \right) \right]$$

(omitting the policy arguments $(a_t \mid s_t)$ for brevity). Now the first use of $\pi_{\theta_{\text{old}}}$ is as part of an importance sampling ratio, for which we must use the behavior policy, and the second and third uses are in applying the implicit KL penalty, for which we can use the proximal policy.

We therefore define the *decoupled clipped objective*

$$L^{\text{CLIP}}_{\text{decoupled}}(\theta) := \hat{\mathbb{E}}_t \left[ \frac{\pi_{\theta_{\text{prox}}}(a_t \mid s_t)}{\pi_{\theta_{\text{behav}}}(a_t \mid s_t)} \min \left( r_t(\theta) \hat{A}_t, \text{clip}(r_t(\theta), 1 - \epsilon, 1 + \epsilon) \hat{A}_t \right) \right],$$

where $\boxed{r_t(\theta) := \dfrac{\pi_\theta(a_t \mid s_t)}{\pi_{\theta_{\text{prox}}}(a_t \mid s_t)}}$.

As a sanity check, note that if we set the KL penalty coefficient $\beta = 0$ or the clipping parameter $\epsilon = \infty$, then the dependence on the proximal policy disappears, and we recover the vanilla (importance-sampled) policy gradient objective (Schulman et al., 2017, equation (6)).

A similar decoupled policy objective is also propsed in Mirror Descent Policy Optimization (Tomar et al., 2020, Section 5.1).

## 3 BATCH SIZE-INVARIANCE

We say an algorithm is *batch size-invariant* to mean that when the batch size is changed, the original behavior can be approximately recovered by adjusting other hyperparameters to compensate. Here we consider behavior as a function of the total number of examples processed, so another way to put this is that doubling the batch size halves the number of steps needed. Shallue et al. (2018) and Zhang et al. (2019) refer to this as "perfect scaling".

We treat batch size-invariance as a descriptive property that can hold to some degree, rather than as a binary property. In practice, the original behavior can never be recovered perfectly, and the extent to which it can be recovered depends on both how much and the direction in which the batch size is changed.

### 3.1 BATCH SIZE-INVARIANCE FOR STOCHASTIC GRADIENT DESCENT

Stochastic gradient descent (SGD) is batch size-invariant, up until the batch size approaches some critical batch size. This is the batch size at which the gradient has a signal-to-noise ratio of around 1. At smaller batch sizes than this, changes to the batch size can be compensated for by a directly proportional adjustment to the learning rate. This core observation has been made many times before (Mandt et al., 2017; Goyal et al., 2017; Smith et al., 2017; Hardin, 2017; Ma et al., 2018; Shallue et al., 2018; McCandlish et al., 2018). A discussion of this and other previous work can be found in Appendix C.

**Sketch explanation.** For the benefit of the reader's intuition, we sketch the explanation for SGD's batch size-invariance. For a much more thorough explanation, we refer the reader to Mandt et al. (2017).

Consider running SGD on a loss function $L(\theta; x)$ of a parameter vector $\theta$ and a data point $x$. Two steps with batch size $n$ and learning rate $\alpha$ corresponds to the update rule

$$\theta_{t+2} = \theta_t - \frac{\alpha}{n} \sum_{x \in B_t} \nabla_\theta L(\theta_t; x) - \frac{\alpha}{n} \sum_{x \in B_{t+1}} \nabla_\theta L(\theta_{t+1}; x),$$

where $B_t$ and $B_{t+1}$ are the next two batches of size $n$. On the other hand, a single step with batch size $2n$ and learning rate $2\alpha$ corresponds to the update rule

$$\theta_{t+2} = \theta_t - \frac{2\alpha}{2n} \sum_{x \in B_t \cup B_{t+1}} \nabla_\theta L(\theta_t; x).$$

These update rules are very similar, the only difference being whether the gradient for $B_{t+1}$ is evaluated at $\theta_t$ or $\theta_{t+1}$. If the batch size is small compared to the critical batch size, then the difference between $\theta_t$ and $\theta_{t+1}$ is mostly noise, and moreover this noise is small compared to the total noise accumulated by $\theta_t$ over previous updates. Hence the two update rules behave very similarly.

A good mental model of SGD in this small-batch regime is of the parameter vector making small, mostly random steps around the loss landscape. Over many steps, the noise is canceled out and the parameter vector gradually moves in the direction of steepest descent. But a single additional step makes almost no difference to gradient evaluations.

In more formal terms, SGD is numerically integrating a stochastic differential equation (SDE). Changing the learning rate in proportion the batch size leaves the SDE unchanged, and only affects the step size of the numerical integration. Once the step size is small enough (the condition that gives

rise to the critical batch size), the discretization error is dominated by the noise, and so the step size stops mattering.

## 3.2 BATCH SIZE-INVARIANCE FOR ADAM

Adam (Kingma & Ba, 2014) is a popular variant of SGD, and is also batch size-invariant until the batch size approaches a critical batch size (which may be different to the critical batch size for SGD) (Zhang et al., 2019). To compensate for the batch size being divided by some constant $c$, one must make the following adjustments (Hardin, 2017):

- Divide the step size $\alpha$ by $\sqrt{c}$.
- (Raise the exponential decay rates $\beta_1$ and $\beta_2$ to the power of $1/c$.)

The first adjustment should be contrasted with the linear learning rate adjustment for vanilla SGD. We discuss the reason for this difference and provide empirical support for the square root rule in Appendix D.

The second adjustment is much less important in practice, since Adam is fairly robust to the $\beta_1$ and $\beta_2$ hyperparameters (hence it has been parenthesized). Note also that $\beta_1$ also affects the relationship between the current policy and the proximal policy in policy optimization. For simplicity, we omitted this adjustment in most of our experiments, but included it in some additional experiments that are detailed in Appendix E.

## 3.3 BATCH SIZE-INVARIANCE FOR POLICY OPTIMIZATION

In policy optimization algorithms like PPO, there are two different batch sizes: the number of environment steps in each gradient step, which we call the *optimization batch size*, and the number of environment steps in each alternation between sampling and optimization, which we call the *iteration batch size*. When we say that such an algorithm is batch size-invariant, we mean that changes to both batch sizes *by the same factor simultaneously* can be compensated for. The motivation for this definition is that this is the effect of changing the degree of data-parallelism.

If the optimization algorithm (such as SGD or Adam) used by PPO is batch size-invariant, then by definition this makes PPO *optimization* batch size-invariant. In the next section, we show how to make PPO *iteration* batch size-invariant, and therefore batch size-invariant outright.

To measure batch size-invariance for policy optimization, there are many features of the algorithm's behavior we could look at. As a simple metric, we use the final performance of the algorithm, since this is the primary quantity of interest to most practitioners. If an algorithm has a high degree of batch size-invariance, then the difference in final performance at different batch sizes should be small.

# 4 PPO-EWMA AND PPG-EWMA

We now introduce a simple modification that can be made to any PPO-based algorithm:

- Maintain an exponentially-weighted moving average (EWMA) of the policy network, updating it after every policy gradient step using some decay rate $\beta_{\mathrm{prox}}$.
- Use this as the network for the proximal policy in one of the decoupled policy objectives.

The motivation for using an EWMA is as follows. We would like to be able to use a policy from some fixed number of steps ago as the proximal policy, but this requires storing a copy of the network from every intermediate step, which is prohibitive if this number of steps is large. Using an EWMA allows us to approximate this policy using more reasonable memory requirements. Although this approximation is not exact, averaging in parameter space may actually improve the proximal policy (Izmailov et al., 2018), and the age of the proximal policy can still be controlled by adjusting $\beta_{\mathrm{prox}}$.

We refer to this modification using the -EWMA suffix. Thus from PPO we obtain PPO-EWMA, and from Phasic Policy Gradient (PPG) (Cobbe et al., 2020) we obtain PPG-EWMA. Pseudocode for PPO-EWMA may be found in Appendix A, and code may be found at [redacted for anonymity].

To see how this modification helps us to achieve batch size-invariance, note that the main effect of changing the iteration batch size in PPO is to change the age of the behavior and proximal policies (which are coupled). The age of the behavior policy affects how on-policy the data is, but this does not matter much, as long as it is not too large. However, the age of the proximal policy affects the strength of the KL penalty (or the implicit KL penalty in the case of the clipped objective), which influences how fast the policy can change. We therefore need to maintain the age of the proximal policy as the iteration batch size is changed, which is what our modification enables.

More specifically, to achieve batch size-invariance for PPO- and PPG-EWMA, we make the following adjustments to compensate for the optimization and iteration batch sizes being divided by some constant $c$:

- Adjust the optimization hyperparameters as described in the previous section, i.e., divide the vanilla SGD learning rate by $c$ or the Adam step size by $\sqrt{c}$. (We use Adam.)
- Multiply $\frac{1}{1-\beta_{\mathrm{prox}}} - 1$ by $c$. (This expression is the center of mass of the proximal policy EWMA, measured in gradient steps.) This adjustment is what keeps the age of the proximal policy constant, measured in environment steps.
- If using advantage normalization, multiply the number of iterations used to estimate the advantage mean variance by $c$. (In practice, we use EWMAs to estimate the mean and variance, and multiply their effective sample sizes, measured in iterations, by $c$.[1]) This keeps the overall sample sizes of these estimates constant, preventing their standard errors becoming too large.
- For PPG, multiply the number of policy iterations per phase $N_\pi$ by $c$. (We use PPG.)

For these adjustments to work, we require that the optimization batch size is sufficiently small. We also require that the number of policy epochs (denoted $E$ in PPO or $E_\pi$ in PPG) is 1. This is because when the iteration batch size is very small, using multiple policy epochs essentially amounts to training on the same data multiple times in a row, which is redundant (modulo changing the learning rate). Our batch size-invariance experiments therefore use PPG-EWMA, where $E_\pi = 1$ is the default.

Note that PPG has a third batch size: the number of environment steps in each alternation between phases, which we call the *phase batch size*. The effect of our adjustment to $N_\pi$ is to simply hold the phase batch size constant, thereby preserving the dynamics of the policy and auxiliary phases.

## 5 Experiments

To validate our analysis, we ran several experiments on Procgen Benchmark (Cobbe et al., 2019), which we found to serve as a useful testbed due to the difficulty and diversity of the environments. Hyperparameters for all of our experiments can be found in Appendix B, and full results on each of the individual environments can be found in Appendix G.

### 5.1 Artificial staleness

To investigate our decoupled policy objectives, we introduced artificial staleness. By this we mean that once data has been sampled through interacting with the environment, it is not immediately used for optimization, but is instead placed in a buffer to be used a fixed number of steps later. Despite being artificial, similar staleness is often encountered in asynchronous training setups, where it is known to cause problems for on-policy algorithms like PPO (OpenAI et al., 2019). We measure staleness in iterations, with one iteration being a single alternation between sampling and optimization.

With artificial staleness, the original PPO objectives are underspecified, since there are two natural choices for $\pi_{\theta_{\mathrm{old}}}$: the policy immediately preceding the current iteration, denoted $\pi_{\theta_{\mathrm{recent}}}$, and the behavior policy $\pi_{\theta_{\mathrm{behav}}}$. However, the decoupled objectives allow us to take the proximal policy $\pi_{\theta_{\mathrm{prox}}}$ to be the recent policy, while continuing to use the behavior policy for importance sampling. This allows the KL penalty (or clipping) to have a consistent effect in terms of controlling how fast the policy changes, while avoiding harmful bias from incorrect importance sampling.

---

[1]The effective sample size, sometimes called the span, of an EWMA with decay rate $\beta$ is equal to $\frac{2}{1-\beta} - 1$.

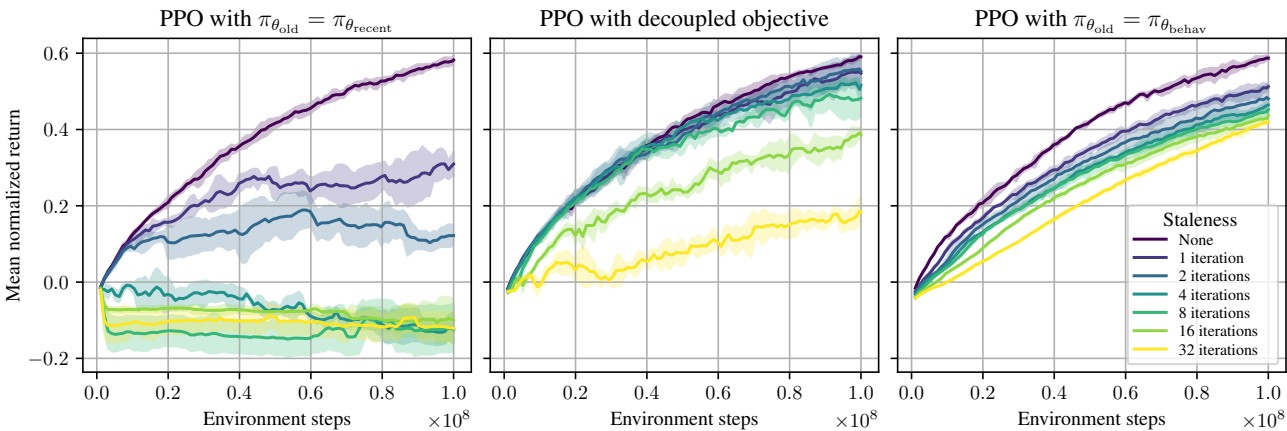

(a) Using the recent policy for importance sampling introduces bias that makes training highly unstable for even small amounts of staleness.

(b) The decoupled objective allows the correct importance sampling ratio to be used while maintaining the age of the proximal policy, preventing performance from degrading much until the data is very stale.

(c) Using the behavior policy to control the size of policy updates holds back learning unnecessarily for small amounts of staleness, but the additional stability is helpful for very stale data.

Figure 1: PPO with artificial staleness, averaged over all 16 Procgen environments. One iteration corresponds to $65,536$ environment steps with our hyperparameters. Mean and standard deviation over 4 seeds shown.

In our experiments, we compare the decoupled objective to both choices for the original objective. Our results are shown in Figure 1. With both choices for the original objective, even a small amount of staleness hurts performance. However, with the decoupled objective, performance is robust to a surprising amount of staleness, with minimal degradation until a staleness of around 8 iterations (over 500,000 environment steps). This demonstrates that the decoupling the proximal policy from the behavior policy can be beneficial.

## 5.2 BATCH SIZE-INVARIANCE

We tested our method of achieving batch size-invariance for PPG-EWMA described in Section 4. Since the optimization batch size is required to be sufficiently small, we started from our default batch size, which uses 256 parallel copies of the environment, and reduced it by factors of 4 until we were running just a single parallel copy of the environment.

Our results are shown in Figures 2 and 3. We were able to achieve a high degree of batch size-invariance, with a difference in final mean normalized return between the largest and smallest batch sizes of 0.052. Moreover, there was a single outlier environment, Heist, without which this difference is reduced to 0.019. We conducted further experiments to try to explain this outlier, which we discuss in Appendix E, but we were not successful.

We conducted ablations in which all but one of the adjustments was removed, the results of which are also shown in Figures 2 and Figure 3. The Adam step size adjustment is the most important at every batch size, and training becomes highly unstable at the smallest batch sizes without this. The advantage normalization adjustment does not matter at the largest batch sizes, but matters a lot at the smallest batch sizes in some environments, which are the ones with particularly noisy advantage standard deviation estimates (see Figure 13 in Appendix G.2). The adjustment to the EWMA matters a little at every batch size, which reflects the fact that PPG is relatively robust to changes to the KL penalty. We did not run an ablation for the adjustment to the PPG hyperparameter $N_\pi$, but can infer from Cobbe et al. (2020, Figure 5) that it comes immediately after the Adam step size adjustment in importance.

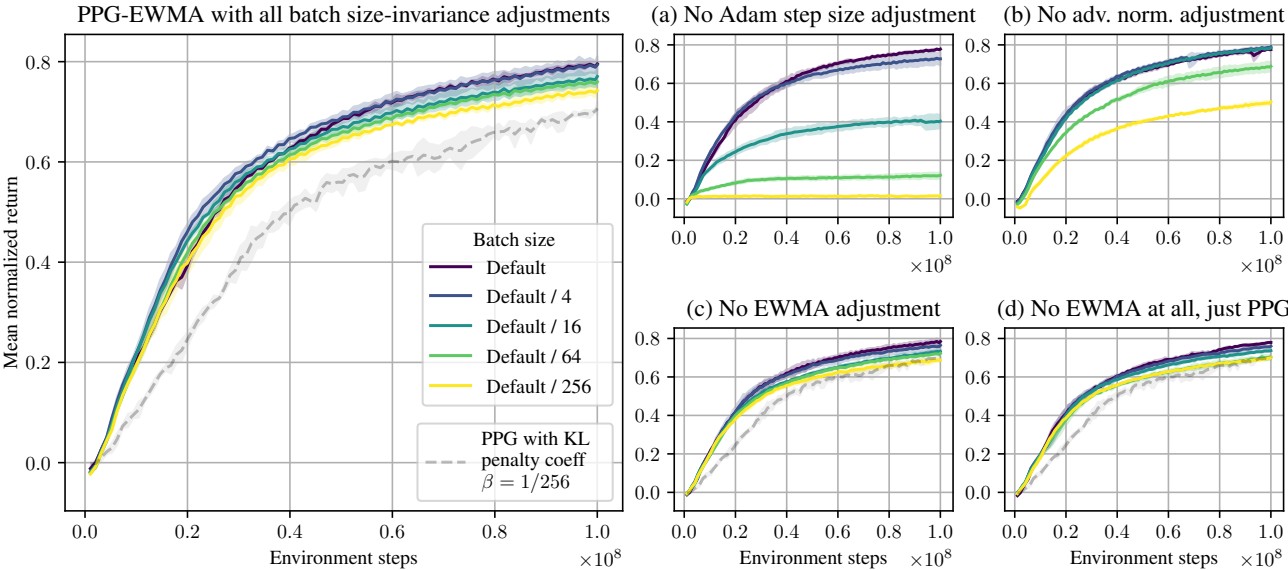

Figure 2: PPG-EWMA at different batch sizes, with hyperparameters adjusted to achieve batch size-invariance, averaged over all 16 Procgen environments. For reference, we also show PPG (at the default batch size) with the KL penalty coefficient ($\beta$ in the $L^{\text{KLPEN}}$ policy objective) reduced to $1/256$, which serves as an approximate lower bound on PPG's performance with a KL penalty that is too weak. On the right we show ablations with all but one of the adjustments. Mean and standard deviation over 3 seeds shown.

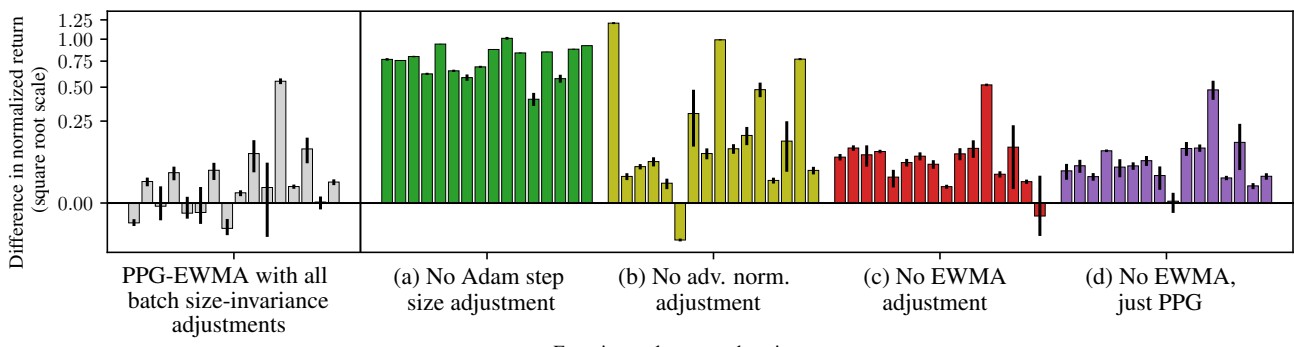

Figure 3: For the results shown in Figure 2, we measure the degree of batch size-invariance for each individual environment by calculating the difference in normalized return between the largest and smallest batch sizes, averaged over the last 4 million timesteps (the length of a single PPG phase). We use a square root scale to make small differences more visible. Mean and standard error over 3 seeds shown.

Environments from left to right: CoinRun, StarPilot, CaveFlyer, Dodgeball, FruitBot, Chaser, Miner, Jumper, Leaper, Maze, BigFish, Heist, Climber, Plunder, Ninja, BossFight.

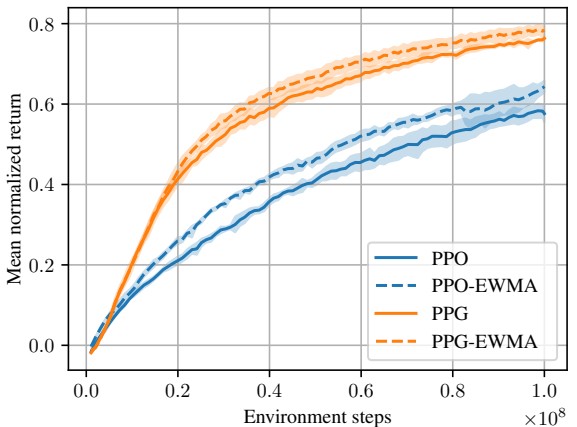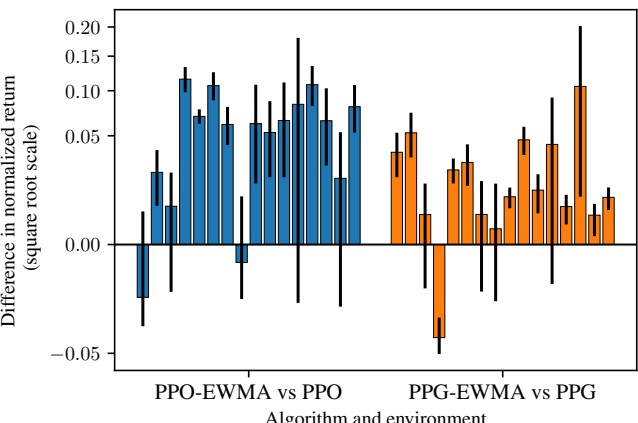

Figure 4: Performance of all 4 algorithms on Procgen. Left: learning curves, mean and standard deviation over 4 seeds shown. Right: difference in normalized return by environment, averaged over the last 4 million timesteps, mean and standard error over 4 seeds shown.

To check the statistical significance of the effects produced by our ablations, we conducted hypothesis tests, which we describe in Appendix F. Our null hypothesis was that the ablation had no effect on the difference in final normalized return at different batch sizes in any of the environments. For the comparison between the largest and smallest batch sizes, we rejected the null hypothesis for all of the ablations at the 0.1% level.

### 5.3 EWMA COMPARISON

Finally, we tested the outright benefit of the EWMA modification by doing a head-to-head comparison of PPO against PPO-EWMA and of PPG against PPG-EWMA. It is important to note that the EWMA introduces an additional hyperparameter $\beta_{\text{prox}}$, but that this was tuned only on the first 8 of the 16 Procgen environments (and only on PPG), and so the algorithms are "complete" on the last 8 environments in the sense of Jordan et al. (2020). Our results are shown in Figure 4.

We found the benefit of the EWMA to be small but remarkably consistent across environments and algorithms, outperforming the baseline on all of the last 8 environments for both PPO and PPG. We believe that this is the result of the EWMA reducing the variance of the proximal policy. Further evidence that the variance of the proximal policy matters is discussed in Appendix H.

Note that this benefit comes at the cost of additional memory to store the weights of the EWMA network, and an additional forward pass of the EWMA network for each policy gradient step. With our hyperparameters, this increases the computational cost of PPO by 30% and of PPG by 2.3%, not including the cost of stepping the environment.[2] [3]

## 6 DISCUSSION

### 6.1 PPO AS A NATURAL POLICY GRADIENT METHOD

Our experiments provide strong empirical support that decoupling the proximal policy from the behavior policy can be beneficial: it can be used to make more efficient use of stale data, to achieve batch size-invariance, and to slightly improve sample efficiency outright. This implies that the usual

---

[2]These costs are calculated as follows. PPO has 1 forward-only and 3 forward-backward passes per environment step, to which PPO-EWMA adds 3 forward-only passes. PPG has 1 forward-only and 7 forward-backward passes of both networks per environment step, to which PPG-EWMA adds 1 forward-only pass of the policy network. A forward-backward pass has 3 times the cost of a forward-only pass.

[3]In practice, including the time taken to step the environment, the EWMA increased wall-clock time by 19% for PPO and by 3% for PPG, but our PPG-EWMA implementation included an additional unnecessary forward pass of an EWMA of the value network.

justification for PPO's surrogate objectives, that they approximate trust region methods, is subtly flawed. Trust region methods keep the policy close to the behavior policy, but it does not matter how far from the behavior policy we move specifically, only that we stay close to *some* recent policy, or in other words, that we do not move too fast. Instead, we speculate that PPO is better viewed as a natural policy gradient method (Kakade, 2001). These methods select updates that *efficiently* improve performance relative to how much the policy is changed.

This conflicts with the results of Schulman et al. (2015), which found constraining updates relative to the behavior policy to be beneficial. With the benefit of hindsight, we believe that at that time, constraint methods had hyperparameters that were easier to tune, but that with the advent of PPO's clipped objective and various normalization schemes, this tends to no longer be the case.

## 6.2 PRACTICAL ADVICE FOR POLICY OPTIMIZATION AT SMALL BATCH SIZES

When solving challenging problems using reinforcement learning, it is often beneficial to increase the batch size to reduce gradient variance. But this is often prohibited by computational resources such as GPU memory, especially with the trend of increasingly large models. The benefit of batch size-invariance is that we can instead train for longer while adjusting other hyperparameters.

However, when working in a new domain, we may need to use a small batch size without knowing which hyperparameters would have worked well at larger batch sizes. We therefore attempt to distill our findings into practical advice for getting policy optimization to work well in a new domain at small batch sizes. Our advice, much of which is already folklore, is as follows:

- By far the most important hyperparameter to tune is the learning rate (or Adam step size). Once it has been tuned for a certain batch size, it can be adjusted formulaically for use at other batch sizes using the rules given in Section 3, as long as the batch size remains small.

- Consider setting the number of policy epochs ($E$ in PPO or $E_\pi$ in PPG) to 1, at least initially. This is the easiest way to maintain stability even if not enough ratios are being clipped. Furthermore, multiple policy epochs are less likely to be beneficial when the iteration batch size is small.

- If using clipping, monitor the fraction of ratios clipped. If it is much less than 1%, then it is probably beneficial to increase the iteration batch size[4], or to use PPO-EWMA with a high $\beta_{\text{prox}}$. If it is much more than 10% with 1 policy epoch or 20% with multiple policy epochs, then this is often a sign that the learning rate is too high.

- If using advantage normalization, monitor the advantage standard deviation estimates. If estimates oscillate by a factor of 10 or more, then it is probably beneficial to perform normalization using data from more iterations.

## 7 CONCLUSION

Policy optimization algorithms such as PPO typically control the size of policy updates using a recent policy we call the proximal policy. We have shown that this policy can be safely decoupled from the behavior policy, which is used to collect experience. We introduced PPO-EWMA and PPG-EWMA, variants of PPO and PPG in which the proximal policy is an exponentially-weighted moving average of the current policy. These variants allow stale data to be used more efficiently, and are slightly more sample efficient outright. Finally, we showed how to make these algorithms batch size-invariant, meaning that when the batch size is changed, we can preserve behavior, as a function of the number of examples processed, by changing other hyperparameters (as long as the batch size is not too large). We discussed our findings, which have both theoretical and practical implications for policy optimization.

---

[4]The iteration batch size can be increased without changing the sampling or optimization batch size by simultaneously increasing the number of timesteps per rollout ($T$) and the number of minibatches per epoch. However, $T$ also affects the amount of bootstrapping performed, and so the GAE bootstrapping parameter ($\lambda$) may also need to be adjusted to compensate.

## 8 ACKNOWLEDGMENTS

We thank the anonymous reviewers for their detailed and thoughtful feedback. [redacted for anonymity]

## 9 REPRODUCIBILITY STATEMENT

All of our experiments can be re-run and all figures re-created using the code at [redacted for anonymity].

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

# A    PSEUDOCODE FOR PPO-EWMA

We provide pseudocode PPO as well as PPO-EWMA to make it clear what changes need to be made:

---

**Algorithm 1** PPO

**for** iteration = $1, 2, \ldots$ **do**
    **for** actor = $1, 2, \ldots, N$ **do**
        Run policy $\pi_{\theta_{\mathrm{old}}}$ in environment
            for $T$ timesteps
        Compute advantage estimates
            $\hat{A}_1, \hat{A}_2, \ldots, \hat{A}_T$
    **end for**
    **for** epoch = $1, 2, \ldots, E$ **do**
        **for** each minibatch **do**

            Optimize objective $L$
                with respect to $\theta$ on minibatch

        **end for**
    **end for**
    $\theta_{\mathrm{old}} \leftarrow \theta$
**end for**

---

**Algorithm 2** PPO-EWMA

**for** iteration = $1, 2, \ldots$ **do**
    **for** actor = $1, 2, \ldots, N$ **do**
        Run policy $\pi_{\theta_{\mathrm{behav}}}$ in environment
            for $T$ timesteps
        Compute advantage estimates
            $\hat{A}_1, \hat{A}_2, \ldots, \hat{A}_T$
    **end for**
    **for** epoch = $1, 2, \ldots, E$ **do**
        **for** each minibatch **do**
            Compute $\pi_{\theta_{\mathrm{prox}}}$ for minibatch
            Optimize objective $L_{\mathrm{decoupled}}$
                with respect to $\theta$ on minibatch
            $\theta_{\mathrm{prox}} \leftarrow \mathrm{EWMA}_{\beta_{\mathrm{prox}}}(\theta)$
        **end for**
    **end for**
    $\theta_{\mathrm{behav}} \leftarrow \theta$
**end for**

---

The expression $\mathrm{EWMA}_{\beta_{\mathrm{prox}}}(\theta)$ is shorthand for

$$\frac{\theta_t + \beta_{\mathrm{prox}}\theta_{t-1} + \beta_{\mathrm{prox}}^2\theta_{t-2} + \cdots + \beta_{\mathrm{prox}}^t\theta_0}{1 + \beta_{\mathrm{prox}} + \beta_{\mathrm{prox}}^2 + \cdots + \beta_{\mathrm{prox}}^t},$$

where $\theta_0, \theta_1, \ldots \theta_t$ are the values of $\theta$ after each gradient step. In practice, we compute this incrementally by initializing $\theta_{\mathrm{prox}} \leftarrow \theta$ and $w \leftarrow 1$, and treating the update $\theta_{\mathrm{prox}} \leftarrow \mathrm{EWMA}_{\beta_{\mathrm{prox}}}(\theta)$ as shorthand for

$$w_{\mathrm{new}} \leftarrow 1 + \beta_{\mathrm{prox}}w$$
$$\theta_{\mathrm{prox}} \leftarrow \frac{1}{w_{\mathrm{new}}}\theta + \beta_{\mathrm{prox}}\frac{w}{w_{\mathrm{new}}}\theta_{\mathrm{prox}}$$
$$w \leftarrow w_{\mathrm{new}}.$$

For PPG-EWMA, we make the same changes to the policy phase, while leaving the auxiliary phase unchanged. However, **the EWMA should be reinitialized at the start of each policy phase**, since $\theta$ changes a lot during the auxiliary phase.

Code for both PPO-EWMA and PPG-EWMA may be found at [redacted for anonymity].

## B  HYPERPARAMETERS

All experiments were on Procgen's hard difficulty, without frame stack, using the convolutional neural network from IMPALA (Espeholt et al., 2018). Unless stated otherwise, experiments lasted for 100 million environment steps.

Table 1: Default hyperparameters shared between PPO and PPG.

| Hyperparameter | Value |
| --- | --- |
| Workers | 4 |
| Parallel environments per worker | 64 |
| Timesteps per rollout ($T$) | 256 |
| Minibatches per epoch | 8 |
| Adam step size ($\alpha$) | $5 \times 10^{-4}$ |
| Value function coefficient | 0.5 |
| Entropy coefficient | 0.01 |
| PPO clipping parameter ($\epsilon$) | 0.2 |
| GAE discount rate ($\gamma$) | 0.999 |
| GAE bootstrapping parameter ($\lambda$) | 0.95 |
| Reward normalization? | Yes |
| Advantage normalization? | Yes |

Table 2: Default PPO-specific hyperparameter.

| Hyperparameter | Value |
| --- | --- |
| Epochs ($E$) | 3 |

Table 3: Default PPG-specific hyperparameters.

| Hyperparameter | Value |
| --- | --- |
| Policy iterations per phase ($N_\pi$) | 32 |
| Policy phase policy epochs ($E_\pi$) | 1 |
| Policy phase value function epochs ($E_V$) | 1 |
| Auxiliary phase epochs ($E_{\mathrm{aux}}$) | 6 |
| Auxiliary phase minibatches per epoch | $16N_\pi$ |
| Auxiliary phase cloning coefficient ($\beta_{\mathrm{clone}}$) | 1 |

For the purpose of the artificial staleness experiments, we clipped $\pi_{\theta_{\mathrm{behav}}}$ to keep the ratio $\frac{\pi_\theta}{\pi_{\theta_{\mathrm{behav}}}}$ below 100, for numerical stability.

For PPG-EWMA, we chose the default EWMA decay rate $\beta_{\mathrm{prox}}$ such that the center of mass of the EWMA, $\frac{1}{1-\beta_{\mathrm{prox}}} - 1$, equaled the number of minibatches per policy phase iteration (8), so that the maximum age of the proximal policy is the same in PPG and PPG-EWMA. We tuned this on the first 8 of the 16 Procgen environments by also trying $\frac{1}{1-\beta_{\mathrm{prox}}} - 1 = 2$ and $\frac{1}{1-\beta_{\mathrm{prox}}} - 1 = 32$, but did not find these to perform better. We did not re-tune $\beta_{\mathrm{prox}}$ on the last 8 Procgen environments or on PPO-EWMA.

Table 4: Default PPO-EWMA and PPG-EWMA specific hyperparameter.

| Hyperparameter | Value |
| --- | --- |
| Proximal policy EWMA decay rate ($\beta_{\mathrm{prox}}$) | 0.889 |

For the batch size-invariance experiments, we made the following changes to the above defaults:

- We reduced the number of parallel environments, first by reducing the number of workers from 4 to 1, and then by reducing the number of parallel environments per worker from 64 to 16 to 4 to 1.

- For the policy phase, we adjusted the Adam step size ($\alpha$), the proximal policy EWMA decay rate ($\beta_{\mathrm{prox}}$), advantage normalization, and the number of policy iterations per phase ($N_\pi$) in the way described in Section 4.

- For the auxiliary phase, we initially tried adjusting the Adam step size in the same way as for the policy phase. This worked well in terms of batch size-invariance, but resulted in prohibitively large wall-clock times at small batch sizes, due to the large number of auxiliary epochs. We therefore simply kept the auxiliary phase minibatch size per worker constant, and only adjusted the Adam step size when reducing the number of workers, not when reducing the number of parallel environments per worker.

## C   PREVIOUS WORK ON BATCH SIZE-INVARIANCE

There has been much previous work on batch size-invariance. The underlying idea of modeling SGD as numerically integrating a stochastic differential equation (SDE) is long-established, going at least as far back as Kushner & Yin (2003). More recently, Mandt et al. (2017) and Hardin (2017) observed that changing the learning rate in proportion the batch size leaves the SDE unchanged, and therefore that SGD is batch size-invariant at small batch sizes. Meanwhile, Goyal et al. (2017) empirically validated this invariance on ImageNet. Smith & Le (2017) and Smith et al. (2017) provided further empirical validation for this, as well as for rules describing how the optimal learning rate changes with momentum and training set size.

The term *critical batch size* for the batch size beyond which SGD is no longer batch size-invariant was introduced by Ma et al. (2018). Since then a number of works have studied how the critical batch size varies between problems, mostly with the motivation of improving training efficiency at large batch sizes (in contrast to our work, which focuses on maintaining performance at small batch sizes). Shallue et al. (2018) studied the effect of architectures, datasets and different forms of momentum on the critical batch size, and introduced the term *perfect scaling* for the regime of batch size-invariance. Golmant et al. (2018) also studied the effect of dataset complexity and size on the critical batch size. McCandlish et al. (2018) measured the critical batch size in a range of domains, and showed that it can be predicted using a measure of the noise-to-signal ratio of the gradient known as the *gradient noise scale*. Finally, Zhang et al. (2019) studied the effect of curvature on the critical batch size using a noisy quadratic model, and showed that preconditioning can be used to increase the critical batch size.

# D   ADAM SQUARE ROOT STEP SIZE ADJUSTMENT

In Section 3, we stated that SGD and Adam have different learning rate adjustment rules. To compensate for the batch size being divided by some constant $c$, one must divide the SGD learning rate by $c$, but divide the Adam step size by $\sqrt{c}$ (Hardin, 2017).

The reason for the difference is that Adam divides the gradient by a running estimate of the root mean square gradient. If the gradient vector at the current step is $g_t$, then this denominator is approximately

$$\sqrt{\mathbb{E}\left[g_t^2\right]} = \sqrt{\mathbb{E}\left[g_t\right]^2 + \mathrm{Var}\left[g_t\right]} = \mathbb{E}\left[g_t\right]\sqrt{1 + \frac{\mathrm{Var}\left[g_t\right]}{E\left[g_t\right]^2}} = \mathbb{E}\left[g_t\right]\sqrt{1 + \frac{\mathcal{B}}{n}},$$

where all operations including the variance operator are applied componentwise, $n$ is the batch size, and $\mathcal{B}$ is a componentwise version of the *gradient noise scale* defined by McCandlish et al. (2018), a measure of the noise-to-signal ratio of the gradient that approximates the critical batch size. Hence if the batch size is small compared to the critical batch size, then $\mathcal{B} \gg n$ for most components, and so the Adam denominator is approximately proportional to $\frac{1}{\sqrt{n}}$.

It follows that if the batch size is divided by some constant $c$, then the Adam denominator is multiplied by approximately $\sqrt{c}$ (providing the batch size is small compared to the critical batch size). Hence Adam is effectively dividing the learning rate by $\sqrt{c}$ automatically, and so the step size $\alpha$ only needs to be adjusted by an additional $\sqrt{c}$ to effectively divide the learning rate by $c$ overall.

This all ignores Adam's $\epsilon$ hyperparameter, which is usually negligible, but is sometimes used to interpolate between Adam and momentum SGD.

To verify the square root rule for Adam, we conducted an ablation of our batch size-invariance experiments, in which we made the exact same adjustments, except that we divided the Adam step size by $c$ instead of by $\sqrt{c}$. Our results are shown in Figure 5. When compared with Figure 2, this clearly shows that the square root rule is superior in our setting. Full results on each of the individual environments can be found in Appendix G.2.

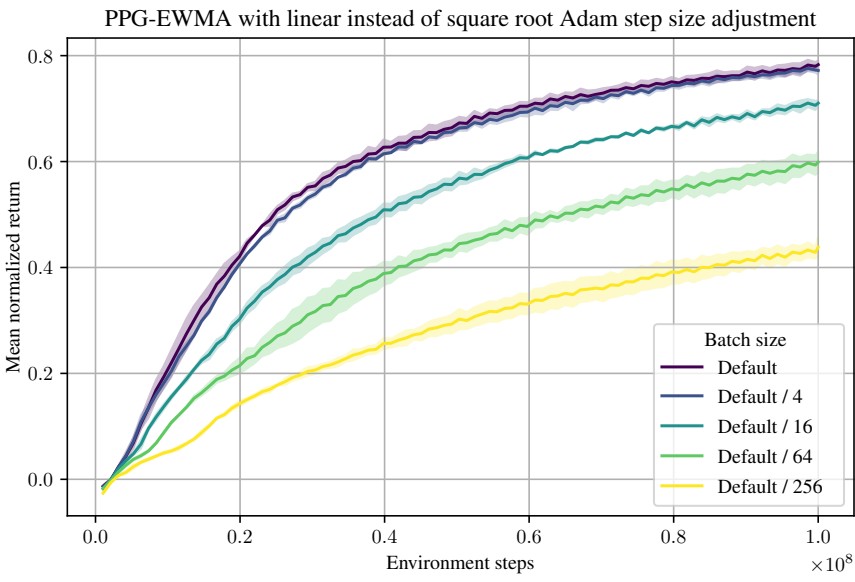

Figure 5: PPG-EWMA at different batch sizes, averaged over all 16 Procgen environments, with hyperparameters adjusted as in Figure 2, except with a linear rather than a square root adjustment to the Adam learning rate. Mean and standard deviation over 3 seeds shown.

Our results are in tension with those of Smith et al. (2017), who verified batch size-invariance for Adam using the linear rather than the square root rule. However, they achieved a lower degree of batch size-invariance with Adam than with SGD (see Figure 4 in that work), and moreover, our results

show that the batch size needs to be reduced significantly before the difference between the two rules is noticeable. We believe that this accounts for their experimental results, and that the square root rule is superior in general (with the exception of when Adam's $\epsilon$ hyperparameter is high enough for it to behave like momentum SGD).

# E  ADAM $\beta_1$ AND $\beta_2$ ADJUSTMENTS

As discussed in Section 3, there is an additional adjustment one should make when using Adam, other than to the step size $\alpha$. To compensate for the batch size being divided by some constant $c$, one should also raise the exponential decay rates $\beta_1$ and $\beta_2$ to the power of $1/c$ (Hardin, 2017).

We omitted this adjustment in most of our experiments, and were still able to achieve a high degree of batch size invariance. For all except one environment, the difference in normalized return between the largest and smallest batch sizes at the end of training was at most 0.11 (see Figure 3). For these environments, it would probably have required many additional experiments to detect any further improvement that adjusting $\beta_1$ and $\beta_2$ might provide. However, for the Heist environment, this difference was 0.55. We hypothesized that this might be explained by the fact that we did not adjust $\beta_1$ and $\beta_2$.

We therefore conducted a version of our batch size-invariance experiments in which we either adjusted only $\beta_2$ using the above rule, or adjusted both $\beta_1$ and $\beta_2$. Our results are shown in Figure 6. In both cases there was still a large difference in performance at the largest and smallest batch sizes.

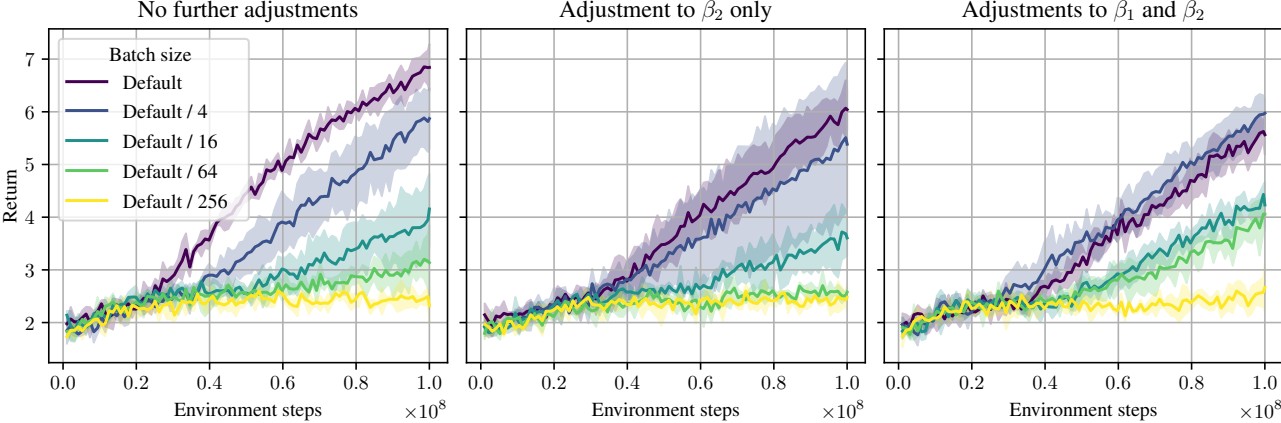

Figure 6: PPG-EWMA at different batch sizes on Heist, with hyperparameters adjusted as in Figure 2, together with further adjustments to Adam's $\beta_1$ and $\beta_2$ hyperparameters as indicated. Mean and standard deviation over 3 seeds shown.

# F  HYPOTHESIS TESTS FOR ABLATIONS

As discussed in Section 3, we conducted hypothesis tests to check the statistical significance of the effects produced by the ablations to our batch size-invariance experiments.

Our primary metric for measuring batch size-invariance was the difference in final performance of the algorithm at different batch sizes. To get a complete picture of how important our ablations were at different batch sizes, we compared our default (largest) batch size with each of the other batch sizes, and tested the hypothesis that the difference was larger for the ablation. This resulted in 16 hypotheses, corresponding to the 4 ablations and the 4 non-default batch sizes. To test each hypothesis, we used a van Elteren test (van Elteren, 1960), a stratified version of the Mann–Whitney $U$-test, treating the different environments as strata. This gives a non-parametric $Z$-test of the null hypothesis that for each of the environments, the probability of the ablation outperforming the original experiment is the same as the probability of the original experiment outperforming the ablation (LaVange & Koch, 2006). To reduce noise (and thereby increase statistical power), we measured average performance over the last 4 million timesteps (the length of a single PPG phase). We used a significance level of 0.1% and applied a Bonferroni correction to account for multiple comparisons.

Our results are shown in Table 5. For ablation (b), the difference is only significant at the smallest batch size. For all other ablations, the difference is significant at every batch size, execpt for the largest batch size for ablation (d).

Table 5: Effect sizes (and $Z$-scores, in parentheses) for each of our hypotheses. The effect size is a difference of differences in final normalized return, between the ablation and the original experiment and between the different batch sizes. In bold are the effect sizes found to be signifcant at the 0.1% level after applying a Bonferroni correction (i.e., with a one-tailed $p$-value below $^{0.001}/_{16}$, or equivalently, a $Z$-score above 3.84).

| Batch sizes: Default vs ... | (a) No Adam step size adjustment | | (b) No adv. norm. adjustment | | (c) No EWMA adjustment | | (d) No EWMA, just PPG | |
|---|---|---|---|---|---|---|---|---|
| Default / 4 | **0.046** | (5.02) | $-0.014$ | $(-1.53)$ | **0.020** | (4.58) | 0.019 | (3.27) |
| Default / 16 | **0.347** | (7.31) | $-0.033$ | $(-1.53)$ | **0.021** | (4.47) | **0.016** | (4.04) |
| Default / 64 | **0.621** | (6.98) | 0.054 | $(-0.11)$ | **0.027** | (5.02) | **0.042** | (5.13) |
| Default / 256 | **0.709** | (6.87) | **0.224** | (4.47) | **0.041** | (4.80) | **0.030** | (4.58) |

# G  RESULTS ON INDIVIDUAL ENVIRONMENTS

## G.1  ARTIFICIAL STALENESS

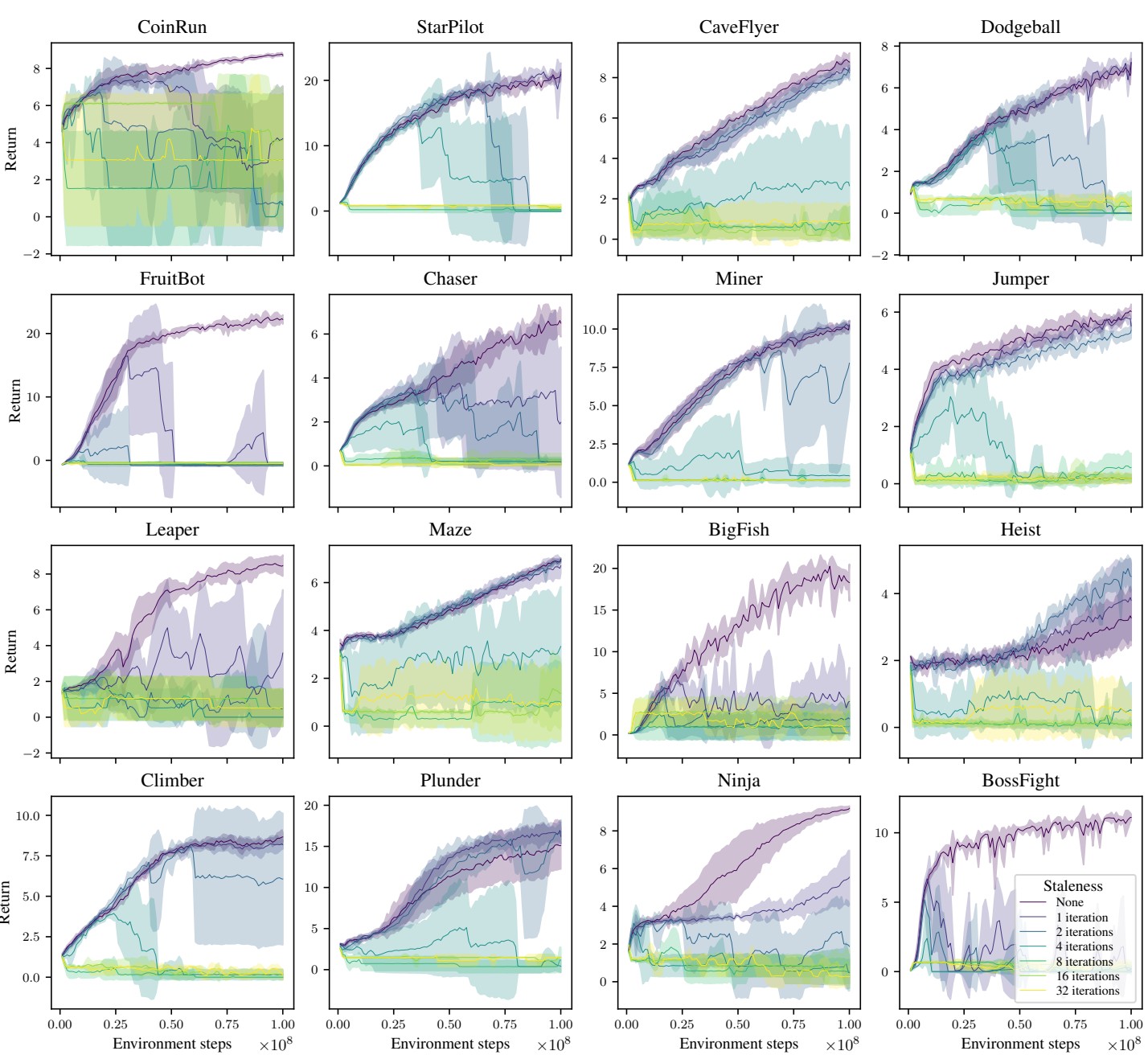

Figure 7: Results from Figure 1(a) (PPO with $\pi_{\theta_{\mathrm{old}}} = \pi_{\theta_{\mathrm{recent}}}$) split across the individual environments. Mean and standard deviation over 4 seeds shown.

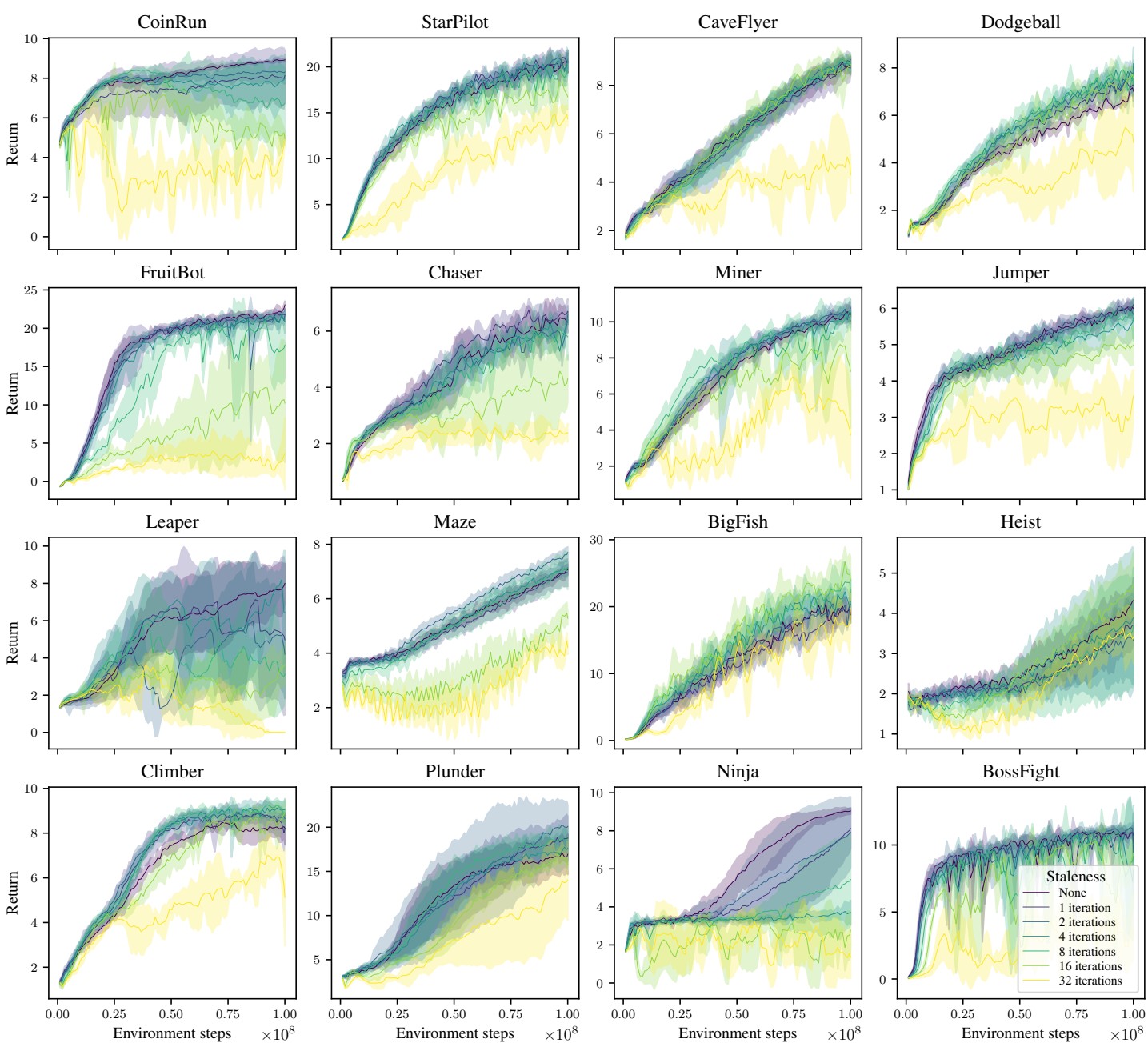

Figure 8: Results from Figure 1(b) (PPO with decoupled objective) split across the individual environments. Mean and standard deviation over 4 seeds shown.

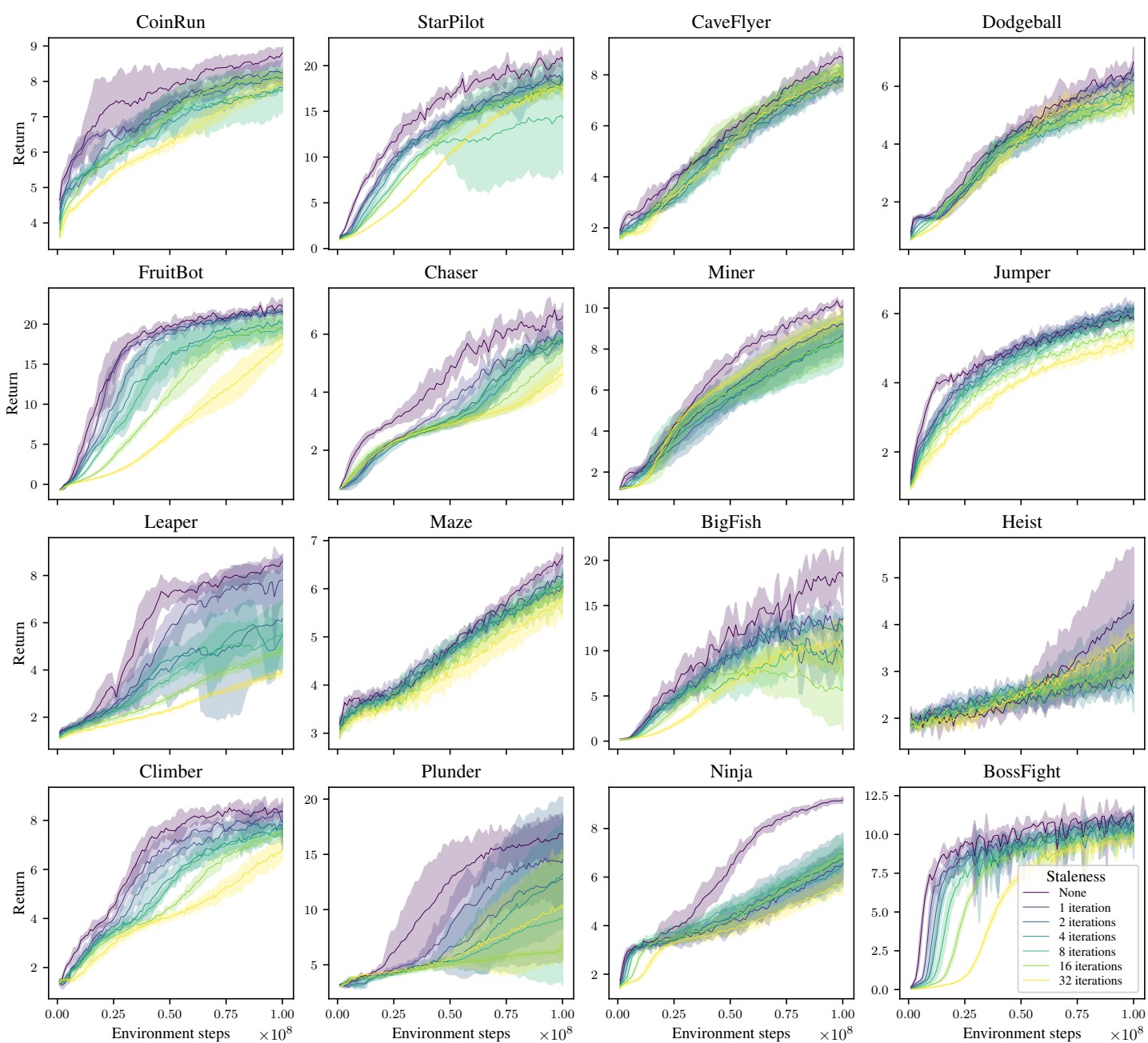

Figure 9: Results from Figure 1(c) (PPO with $\pi_{\theta_{\mathrm{old}}} = \pi_{\theta_{\mathrm{behav}}}$) split across the individual environments. Mean and standard deviation over 4 seeds shown.

## G.2 BATCH SIZE-INVARIANCE

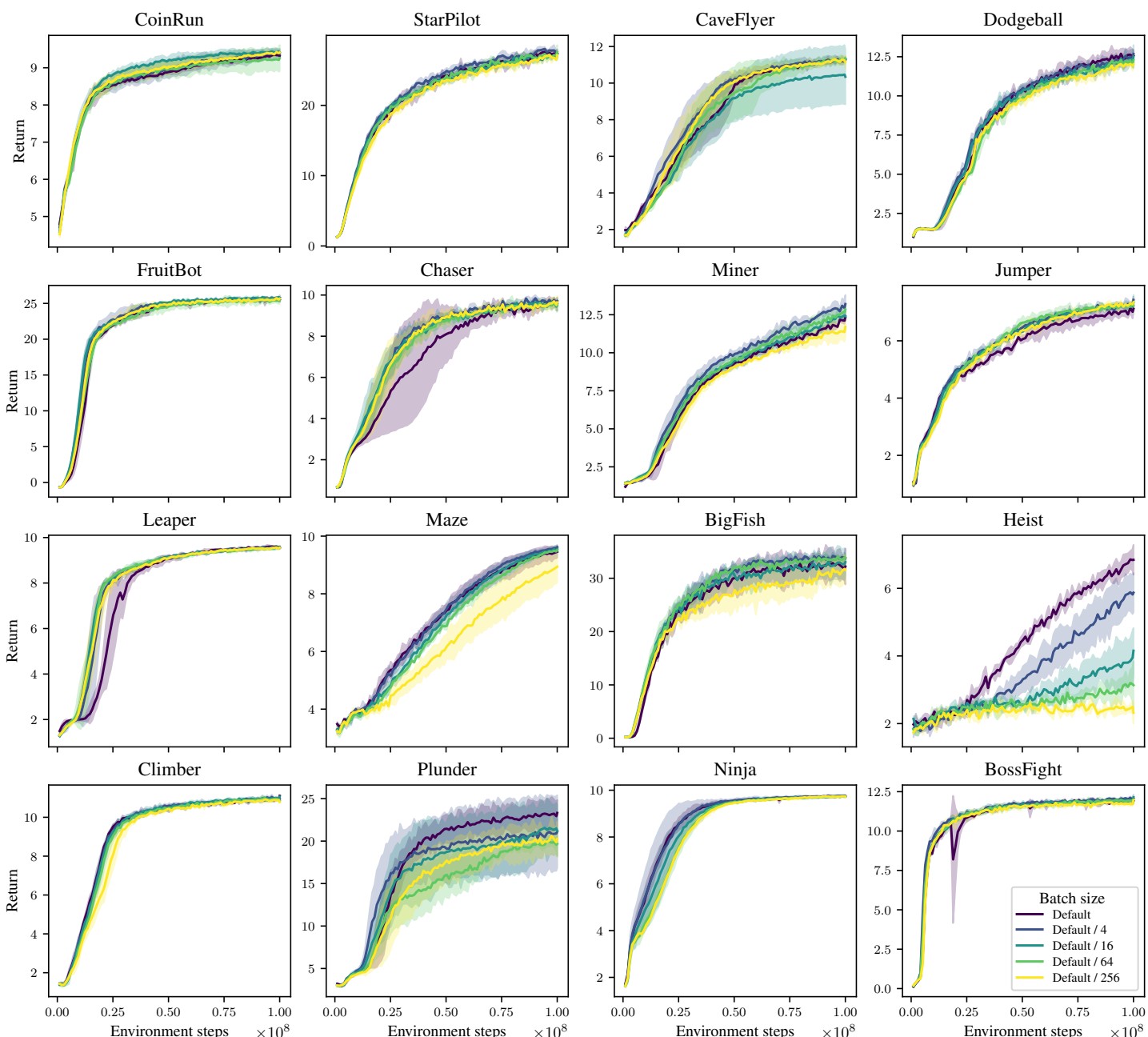

Figure 10: Results from Figure 2 (PPG-EWMA with all batch size-invariance adjustments) split across the individual environments. Mean and standard deviation over 3 seeds shown.

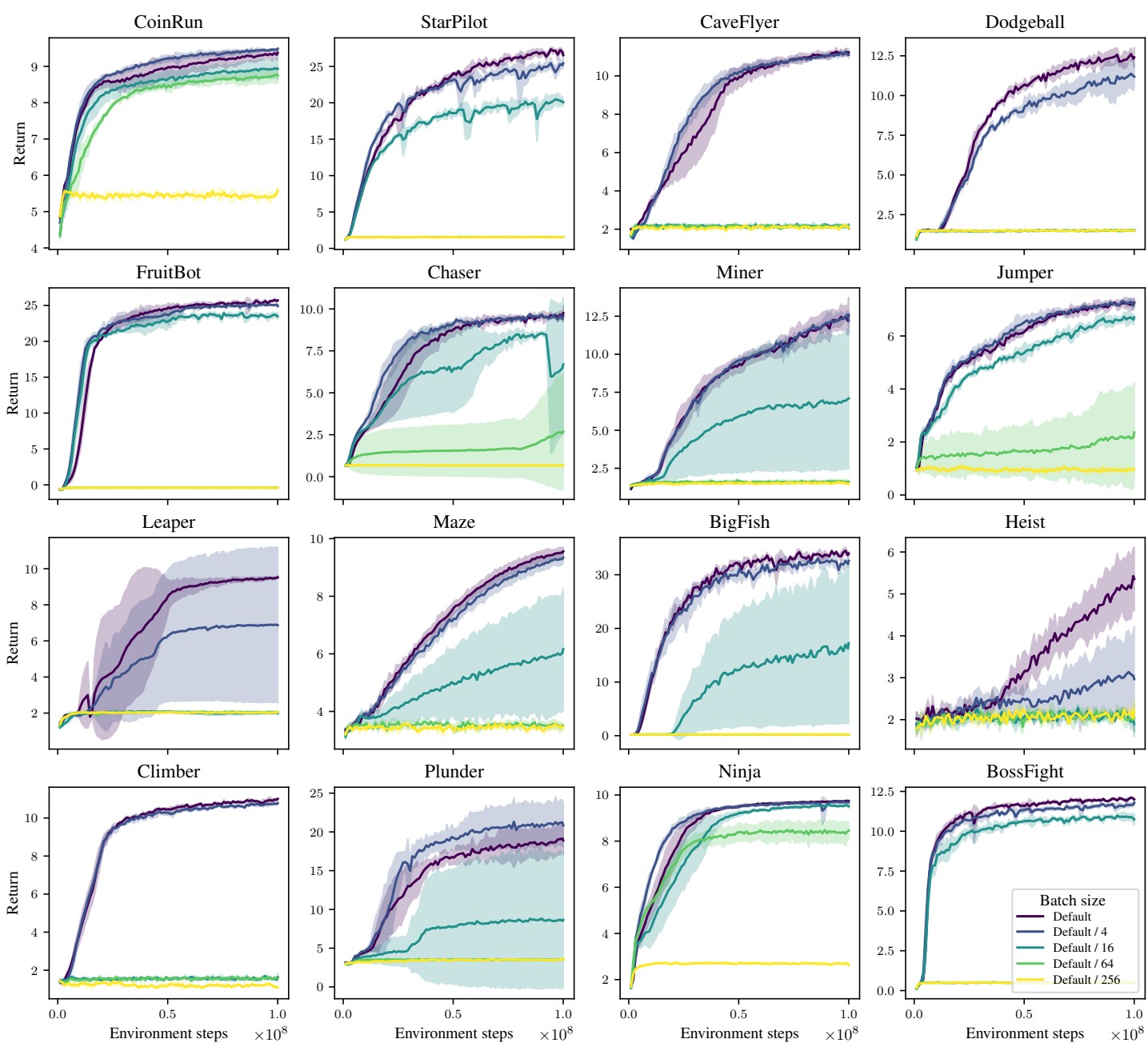

Figure 11: Results from Figure 2(a) (no Adam step size adjustment) split across the individual environments. Mean and standard deviation over 3 seeds shown.

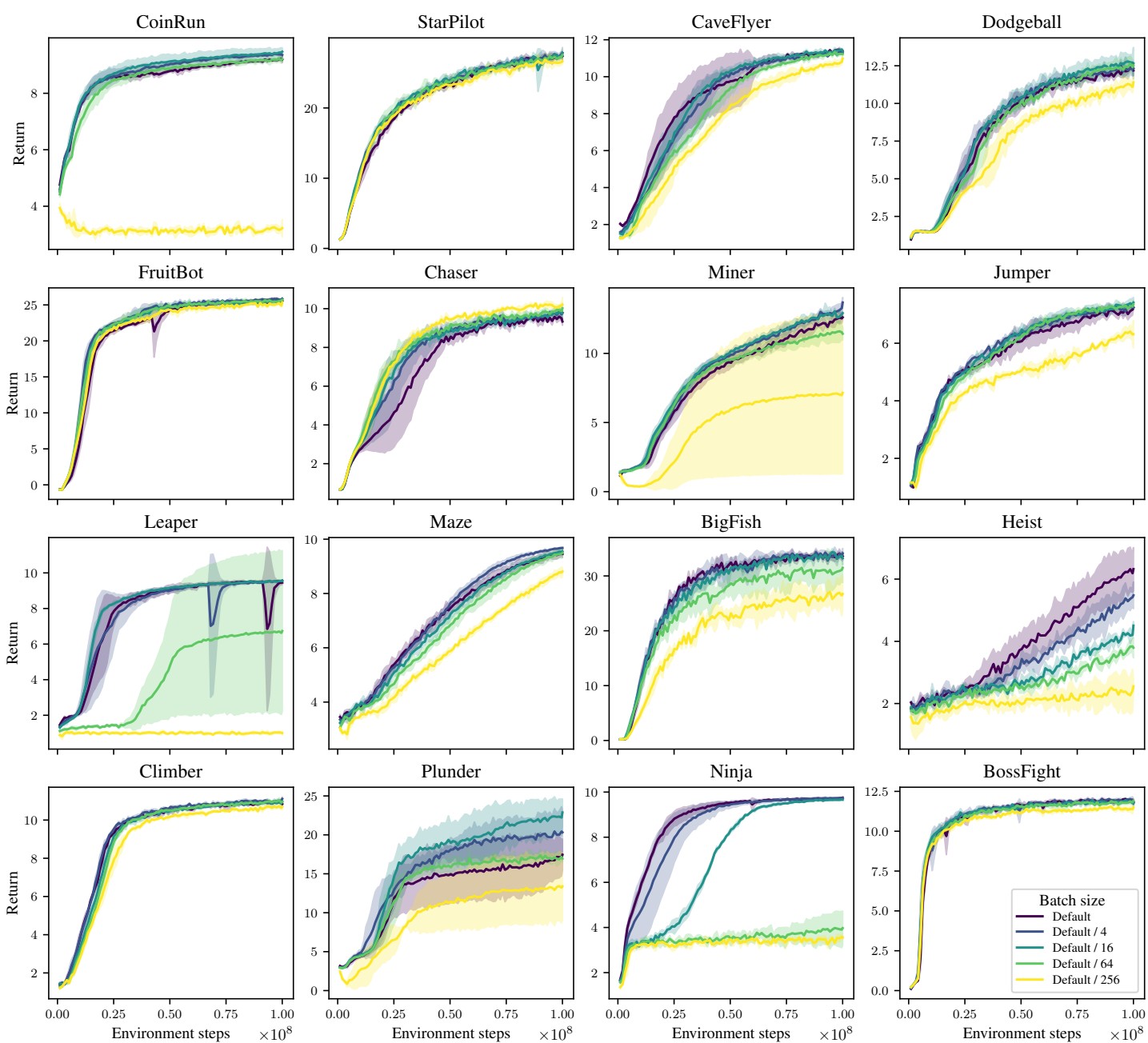

Figure 12: Results from Figure 2(b) (no advantage normalization adjustment) split across the individual environments. Mean and standard deviation over 3 seeds shown.

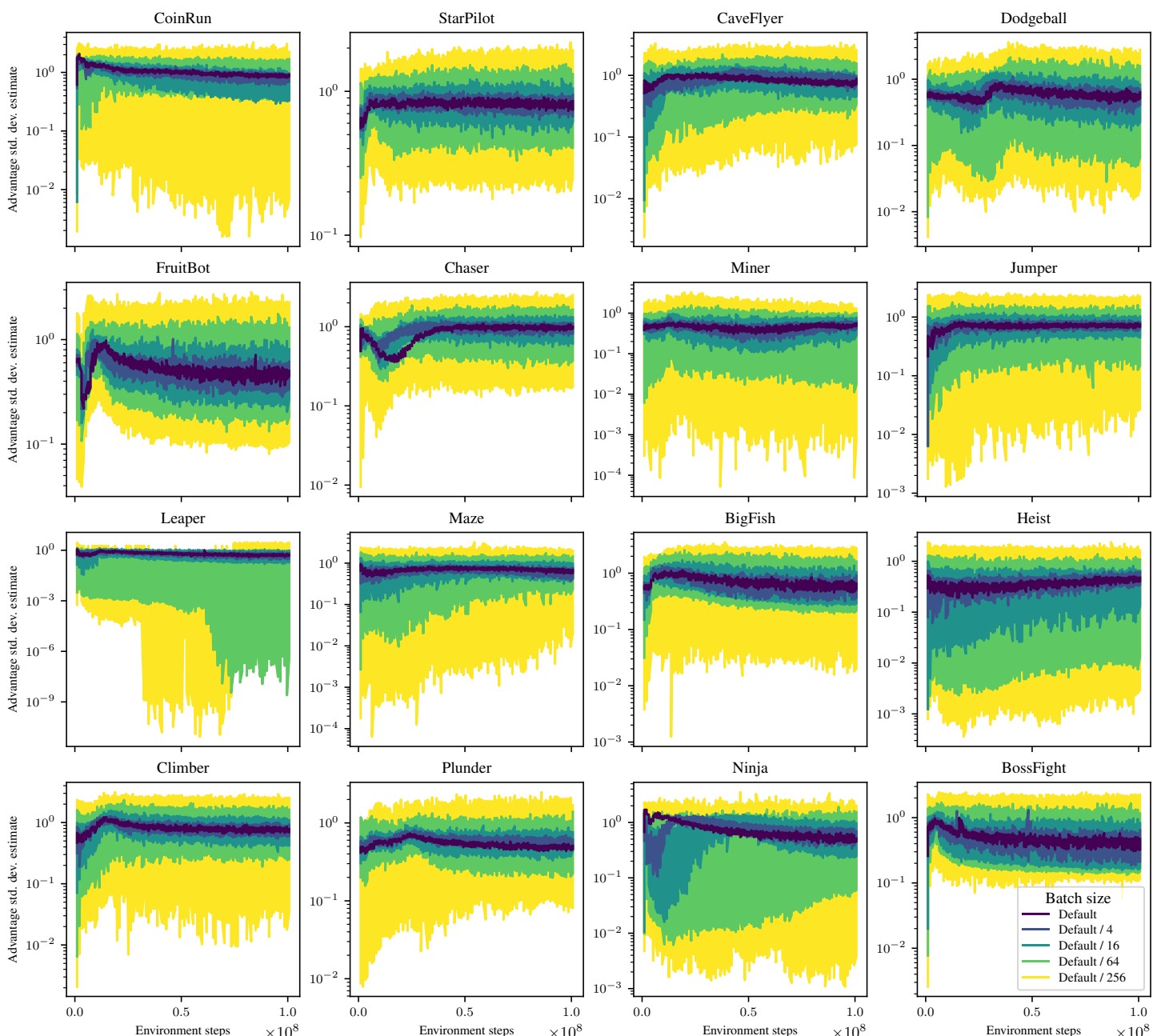

Figure 13: Advantage standard deviation estimates for the results from the previous figure. We plot estimates from the first seed only and perform no smoothing, since we are interested in the amount of oscillation. Note that performance degrades with no advantage normalization adjustment only once the estimates oscillate by factor of around 10 or more.

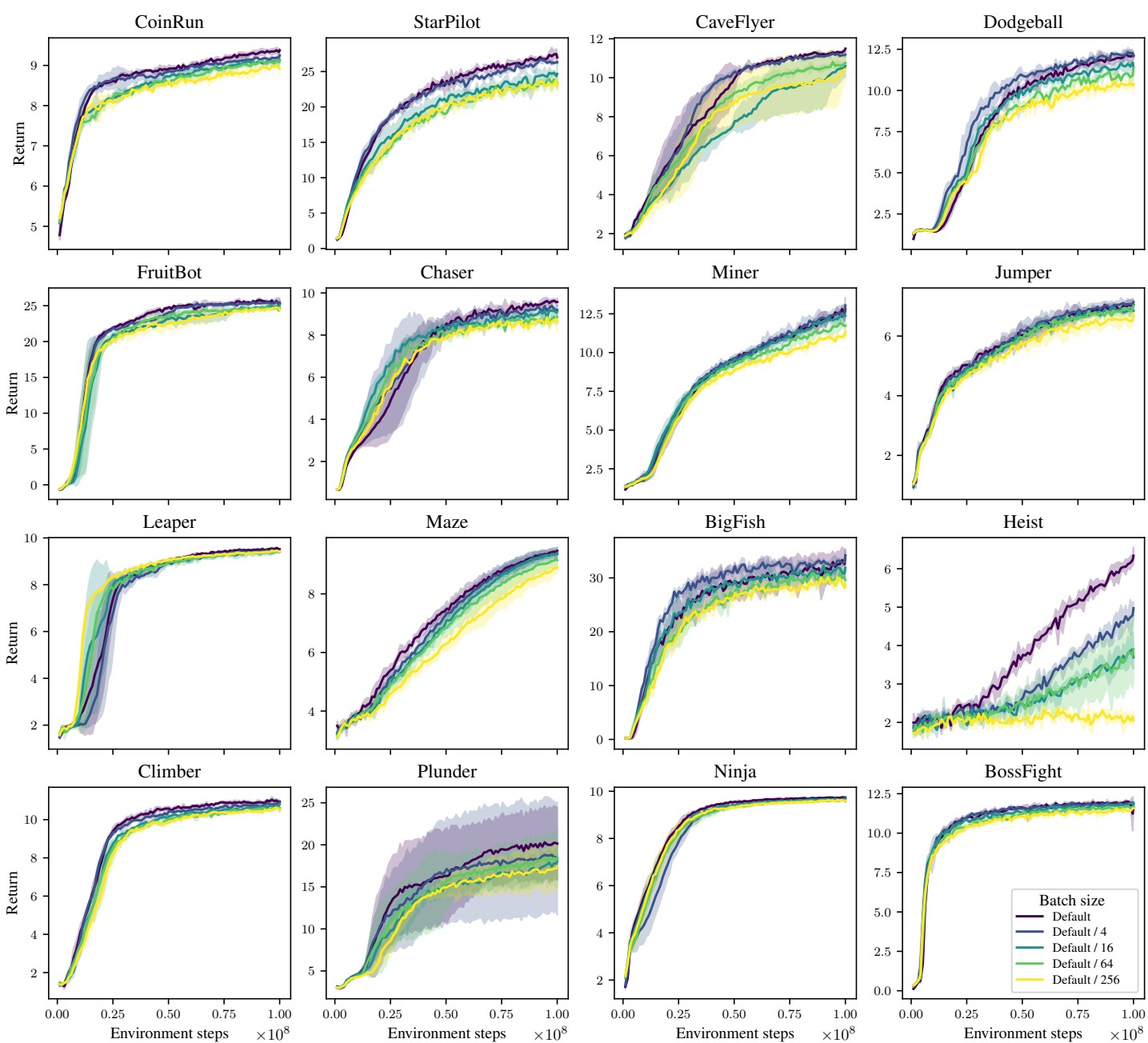

Figure 14: Results from Figure 2(c) (no EWMA adjustment) split across the individual environments. Mean and standard deviation over 3 seeds shown.

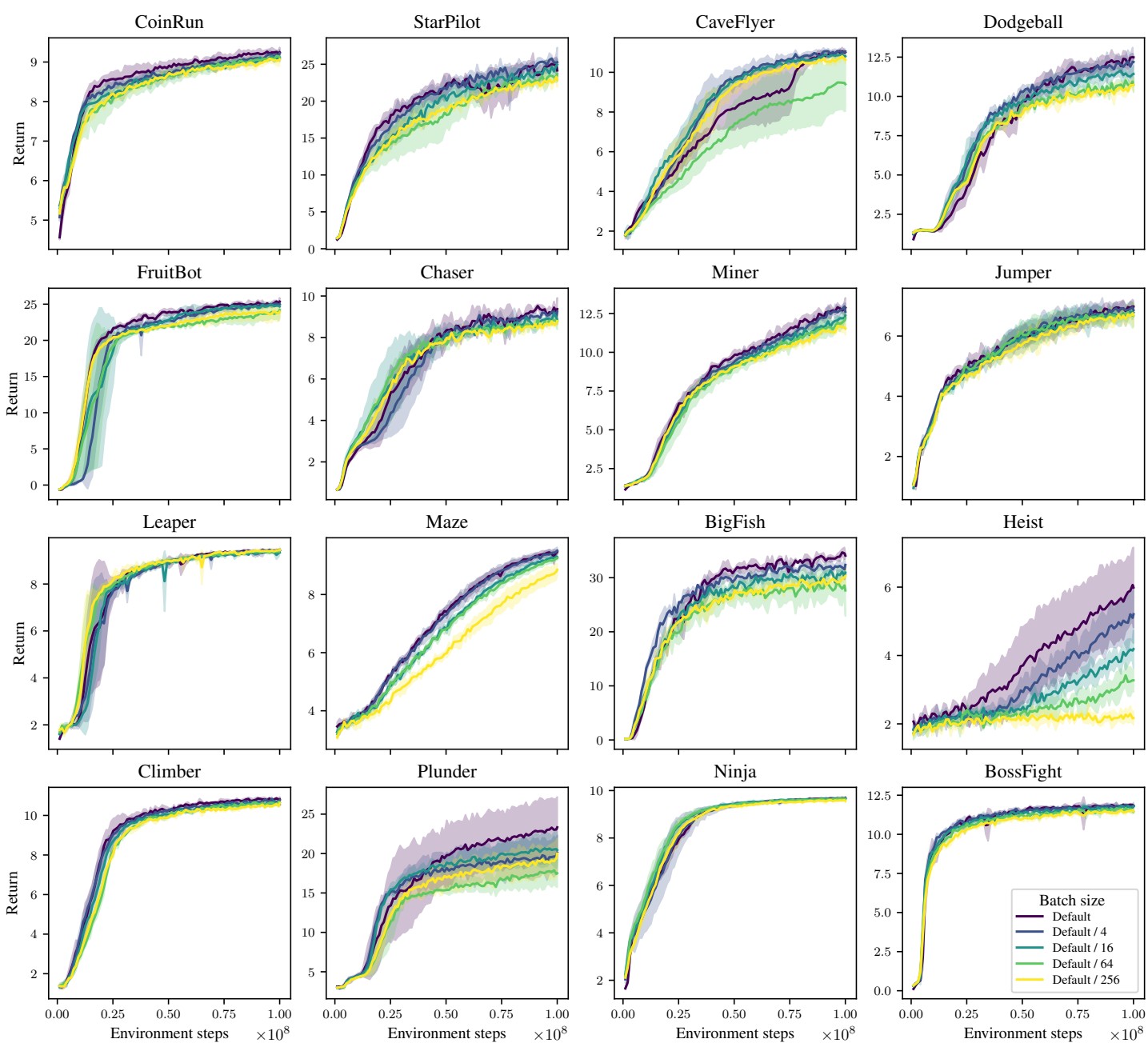

Figure 15: Results from Figure 2(d) (no EWMA at all, just PPG) split across the individual environments. Mean and standard deviation over 3 seeds shown.

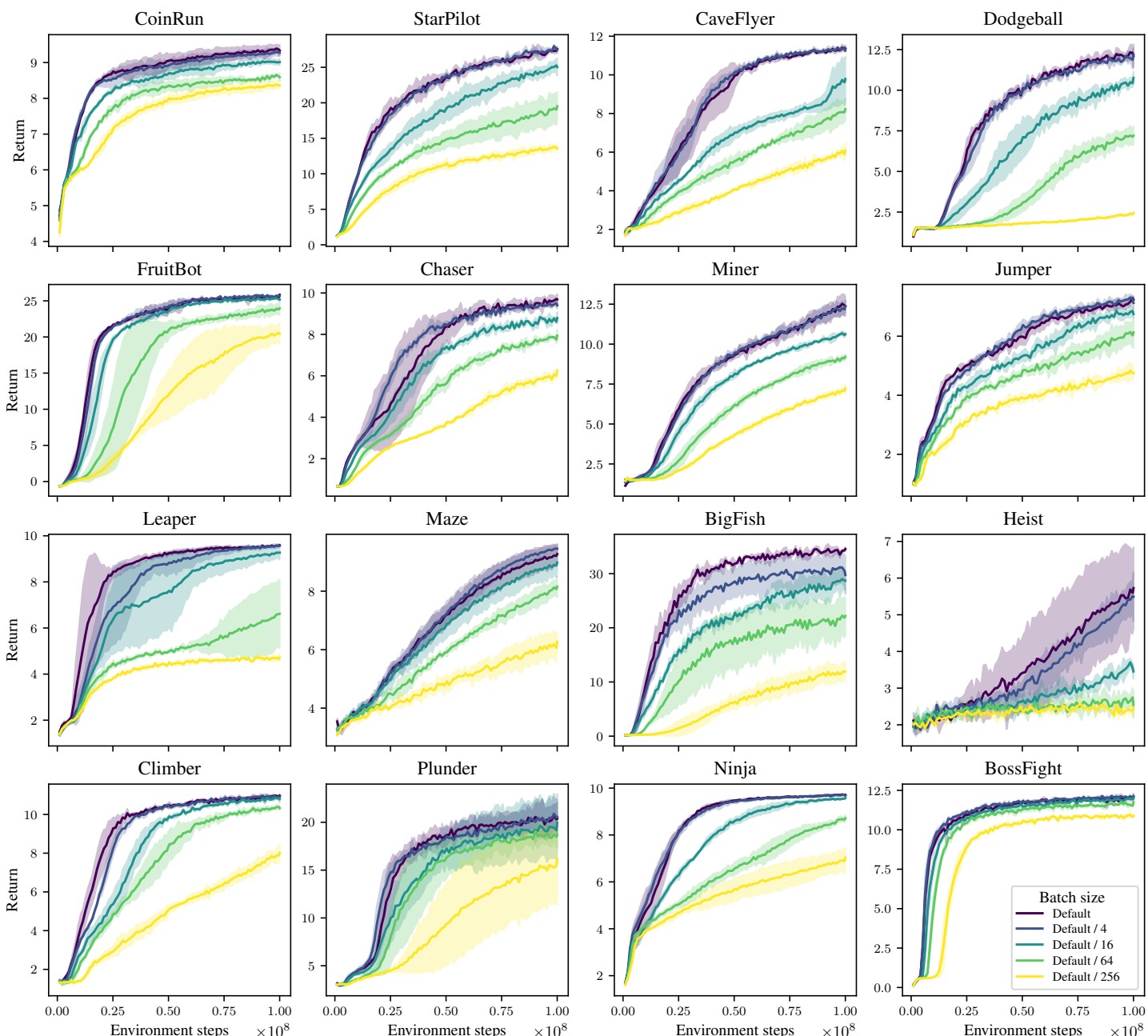

Figure 16: Results from Figure 5 (PPG-EWMA with linear instead of square root Adam step size adjustment) split across the individual environments. Mean and standard deviation over 3 seeds shown.

## G.3 EWMA COMPARISON

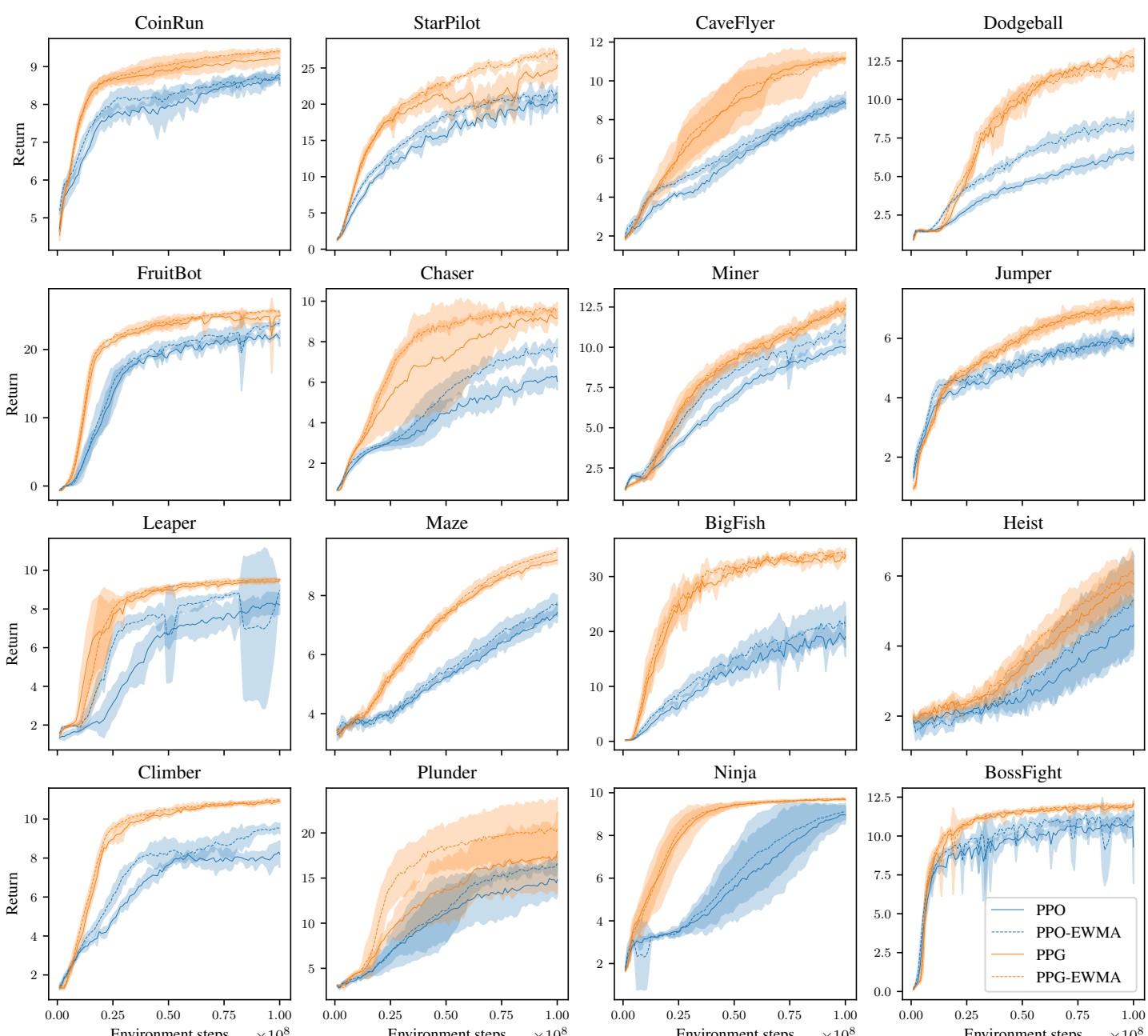

Figure 17: Results from Figure 4 (performance of all 4 algorithms) split across the individual environments. Mean and standard deviation over 4 seeds shown.

## H    ROLE OF THE PROXIMAL POLICY EWMA DECAY RATE

In this section we discuss the role of the hyperparameter $\beta_{\text{prox}}$ in PPO-EWMA and PPG-EWMA in more depth.

Recall that in PPO-EWMA and PPG-EWMA, the proximal policy network parameter vector $\theta_{\text{prox}}$ is an exponentially-weighted moving average (EWMA) of the policy network parameter vector $\theta$, meaning that

$$\theta_{\text{prox}} = \frac{\theta_t + \beta_{\text{prox}}\theta_{t-1} + \beta_{\text{prox}}^2\theta_{t-2} + \cdots + \beta_{\text{prox}}^t\theta_0}{1 \phantom{} + \beta_{\text{prox}} \phantom{aaa} + \beta_{\text{prox}}^2 \phantom{aaa} + \cdots + \beta_{\text{prox}}^t},$$

where $\theta_0, \theta_1, \ldots \theta_t$ are the values of $\theta$ after each gradient step.

Rather than working with the decay rate $\beta_{\text{prox}}$ of this EWMA directly, it is conceptually clearer to work with the center of mass of this EWMA,

$$\text{COM}_{\text{prox}} := \lim_{t \to \infty} \frac{0 + \beta_{\text{prox}}1 + \beta_{\text{prox}}^2 2 + \cdots + \beta_{\text{prox}}^t t}{1 + \beta_{\text{prox}} \phantom{} + \beta_{\text{prox}}^2 \phantom{} + \cdots + \beta_{\text{prox}}^t} = \frac{1}{1 - \beta_{\text{prox}}} - 1.$$

This is average age of a term in the EWMA in the limit as $t \to \infty$, and so if $\theta$ were to follow a straight line path for example, then $\theta - \theta_{\text{prox}}$ would be approximately proportional to $\text{COM}_{\text{prox}}$.

Suppose then that we halve $\text{COM}_{\text{prox}}$. What is the effect of this?

Consider the gradient of the KL divergence from the current policy to the proximal policy, as a function of the proximal policy parameter vector,

$$\text{Grad-KL}\left(\theta_{\text{prox}}\right) := \nabla_\theta \text{KL}\left[\pi_{\theta_{\text{prox}}}\left(\cdot \mid s_t\right), \pi_\theta\left(\cdot \mid s_t\right)\right].$$

Since KL divergence is always greater than or equal to 0, with equality if and only if the input distributions are equal, $\text{Grad-KL}\left(\theta\right) = 0$, and hence, to first-order, $\text{Grad-KL}\left(\theta_{\text{prox}}\right)$ is a linear function of $\theta - \theta_{\text{prox}}$. Therefore halving $\text{COM}_{\text{prox}}$ should have a similar effect to halving the KL penalty coefficient $\beta$. In other words, we should be able to compensate for halving $\text{COM}_{\text{prox}}$ by doubling the KL penalty coefficient.

Intuitively, the KL penalty acts like a rubber band pulling the policy towards the proximal policy. Halving $\text{COM}_{\text{prox}}$ is analogous to attaching the rubber band to a point half as far away, while doubling the KL penalty coefficient is analogous to doubling the thickness of the rubber band. Doing both simultaneously results in the same overall force.

We tested this hypothesis using a hyperparameter grid search over the EWMA center of mass and the KL penalty coefficient, for PPG-EWMA on StarPilot. We used our smallest batch size, along with our corresponding batch size-invariance adjustments, to allow the greatest scope for reducing $C_{\text{prox}}$ without making the EWMA degenerate into averaging over a single data point.

Our results are shown in Figure 18. The diagonal banding clearly demonstrates the expected effect. However, the effect only holds locally: as $\text{COM}_{\text{prox}}$ is continually halved and the KL penalty coefficient is continually doubled, performance gradually degrades.

We believe that this is because reducing $\text{COM}_{\text{prox}}$ has a second-order effect, which is to increase the variance of $\theta - \theta_{\text{prox}}$. This is both because the EWMA is averaging over a smaller effective sample size, and because $\theta_t - \theta_{t-k}$ has a lower signal-to-noise ratio as $k$ decreases. Therefore if $\text{COM}_{\text{prox}}$ has been halved too many times, we should expect to no longer be able to fully compensate for this by continuing to double the KL penalty coefficient.

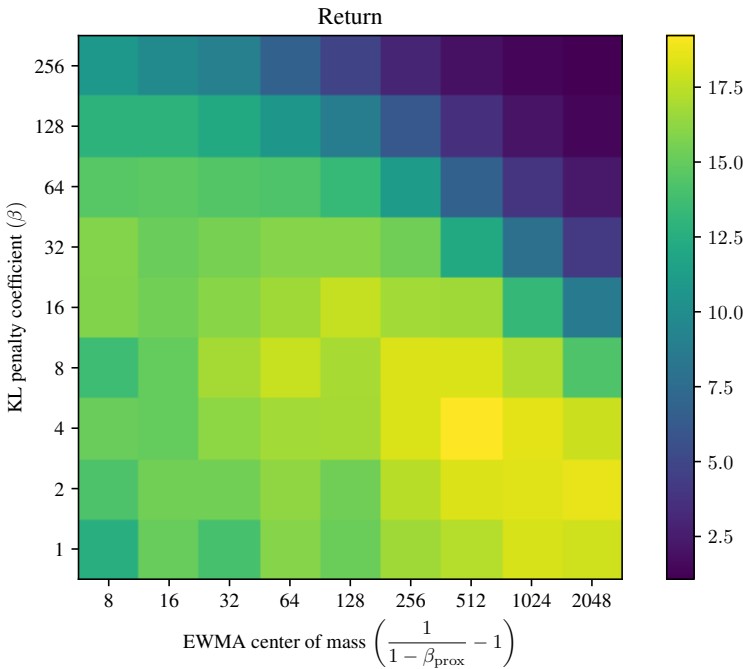

Figure 18: Performance on PPG-EWMA on StarPilot after 20 million environment timesteps, using a single parallel copy of the environment along with our batch size-invariance adjustments. The default hyperparameter settings correspond to square in the bottom right corner. Mean over 2 seeds shown.

