# OpenReview forum: "Batch size-invariance for policy optimization"
_ICLR.cc/2022/Conference — ICLR 2022 Submitted_

### Official Review · Reviewer_pfgK · 2021-10-31

**Correctness:** 2
**Technical Novelty And Significance:** 2
**Empirical Novelty And Significance:** 2
**Recommendation:** 5
**Confidence:** 4

**Main Review:**


Strengths:

This paper study an interesting problem of batch size-invariant.
The writing is clear, and the method is easy to understand.
The experimental results are sufficient to demonstrate the effectiveness of the proposed method.



Weaknesses:
- The paper does not provide any formal theoretical analysis of the batch size-invariant property. Instead, the description is very vague (e.g., Sec 3.1 Sec 4)."Hence the two update rules behave very similarly." To what extent do they behave similarly? Can you give a formal analysis?
- Using moving average policy (instead of behavior policy) as proxy policy $\pi_{proxy}$ is interesting. However, this method seems not to be theoretical-justified. What is the motivation of averaging over parameter space? Is averaging over parameter space equivalent to averaging over policy space? Moreover, does this method still enjoy the monotonical improvement property (as in [1])?

[1] Liu B, Cai Q, Yang Z, et al. Neural proximal/trust region policy optimization attains globally optimal policy[J]. Advances in Neural Information Processing Systems, 2019, 32.


**Summary Of The Paper:**

This paper studies the method to achieve the batch size-invariant for policy gradient algorithms (PPO, PPG). The paper achieves this by decoupling the proximal policy from the behavior policy. The experiments demonstrate the effectiveness of the method.


**Summary Of The Review:**

This paper presented some interesting ideas of batch size-invariance of policy gradient.
Although the experimental results well demonstrated the effectiveness of their method.
The method lacks theoretical justification, and the claim of the batch size-invariant property is vague and weak.

---

> ### Author Response · Authors · 2021-11-17
> **Response to Reviewer pfgK**
>
> We are truly grateful to the reviewer for their balanced and constructive review.
>
> The reviewer identified two high-level weaknesses in our work. Firstly, many of our claims were vague or unsupported. We have now released an updated version of our work that is greatly improved on this front. The most important change is the introduction of a metric that allows us to measure the degree of batch size-invariance, allowing us to make more precise claims. In particular, we use it to provide a more rigorous analysis of our ablations, showing that the effects of all of them are statistically significant.
>
> We have made many additional improvements along similar lines in response to the concerns of reviewer tiiD, such as removing all mentions of "complete" batch size-invariance, and instead referring to the degree of batch size-invariance, which is measured by our new metric.
>
> Secondly, the reviewer felt that our method lacked theoretical justification, especially the use of the EWMA. It is true that the EWMA was motivated on practical grounds: we wished to approximate a policy from a fixed number of steps ago without storing a copy of the network for every intermediate step. Still, averaging in parameter space is not without precedent - see for example Izmailov et. al. (2018), which shows that it leads to better generalization.
>
> In response to the specific weaknesses identified:
> - "Instead, the description is very vague (e.g., Sec 3.1)" - The claims in this section are well-established in the literature, and our explanation was only intended to serve as a sketch for the benefit of the reader's intuition. However, we did not make this clear enough. We have now clearly signposted this explanation as a sketch and referred the reader to Mandt et. al. (2017) for a more thorough explanation.
> - "What is the motivation of averaging over parameter space?" - We have added more detail about the practical motivation for this described above. However, we have not given a precise theoretical motivation. This is because this choice was in fact made using practical heuristics, and so we think it would be misleading to claim that it was closely guided by a theoretical result.
>
> Although our work does not contain many theoretical results, we hope that the variety of empirical phenomena studied will provide fertile ground for future theoretical research, helping to bridge the gap between theory and practice.
>
> Pavel Izmailov, Dmitrii Podoprikhin, Timur Garipov, Dmitry Vetrov, and Andrew Gordon Wilson. "Averaging weights leads to wider optima and better generalization." arXiv preprint arXiv:1803.05407, 2018.
>
> Stephan Mandt, Matthew D Hoffman, and David M Blei. "Stochastic gradient descent as approximate Bayesian inference." arXiv preprint arXiv:1704.04289, 2017.

---

> > ### Comment · Reviewer_pfgK · 2021-11-29
> > **Review after response**
> >
> > I want to thank their response and their efforts on my concerns.
> >
> > I agree with reviewer kR2g that Figure 3 is very interesting.
> > The author also provides a new metric to measure the degree of batch size-invariance. I appreciate the efforts. However, it is not very convincing as there are many randomnesses that can affect the final performance (and the 3 random seeds is either not very convincing). I expect to see some more solid and convincing measurement and theoretical analysis.
> >
> > I think there are some improvements with the paper.
> >  My suggestion is to reject it.
> > I hope the authors improve the paper for submission to the next venue using these reviews.

---

> > > ### Author Response · Authors · 2021-11-29
> > > **Follow-up response**
> > >
> > > We are very grateful to the reviewer for their further comments. However, we respectfully disagree that our results could reasonably be the result of random chance. We would like to check that the reviewer did not miss addition of Appendix F, where we addressed this concern by conducting hypothesis tests. These show that the effects of all of our ablations are statistically significant at the 0.1% level. Although we only used 3 random seeds, the large number of environments used gives us enough statistical power to make this strong claim of significance.

---

> > > > ### Comment · Reviewer_pfgK · 2021-11-30
> > > > **Thanks for the authors' response**
> > > >
> > > > Thanks for your explanation.
> > > >
> > > > I believe the result can demonstrate that your method can be batch-invariant (and I never suspect it both  in my initial review and my response). I appreciate the additional results in Figure 3 and Appendix F, which did improve my confidence.
> > > >
> > > > However, I believe there is still space for improvements of the paper, e.g., the theoretical analysis on how this method could be batch-invariant in the surrogate loss or final performance.  I encourage the authors to polish and resubmit the paper, hoping to see it in a future conference.

---

### Official Review · Reviewer_kR2g · 2021-11-01

**Correctness:** 3
**Technical Novelty And Significance:** 2
**Empirical Novelty And Significance:** 2
**Recommendation:** 8
**Confidence:** 5

**Main Review:**

Firstly, the paper is clearly written, is simple to read, and flows well. I particularly like that the authors do not oversell any of the results. In terms of motivation, batch size invariance is an important and interesting property and insights specific to RL could be useful for future research.

On the other hand, I do not understand the main takeaway from the paper. There are two clear contributions here, one of the decoupling objective and the other of achieving batch size invariance. Currently, I am unable to place either of these as the main objective/problem of the paper. If it's the decoupling, then it should lead to better batch size invariance than the standard, coupled policy agent. I think this is plotted in Figure 2 (d), where the coupled PPG agent is used with all optimization adjustments but achieves similar batch invariance as the decoupled PPG agent as in Figure 2. So it seems like batch invariance can be achieved irrespective of the algorithm, i.e. if it's coupled or decoupled. Furthermore, as it is, the decopling only leads to a marginal improvement over the coupled agent. Similarly, if batch size invariance is the main objective, then I don't see how the decoupling idea is worth looking at. Any comments from the authors on this would be really helpful.

Minor Comments/Questions:

1. Typo at line 4 of section 3.1.
2. Last line of section 3.1 "Once the step size is small enough...": Does this small enough batch size have anything to do with the critical batch size? I assume not right?
3. Last line of section 6.1 "We speculate that updates should improve...": Are you trying to say that more the policy differs a lot from the behavior policy, the more performance improvement we'll see? I do not completely understand why the conclusion here is different from that of the PPO paper.
4. Second paragraph of section 6.2: The first sentence talks about large batch sizes but the next line mentions advice regarding small batch sizes. They seem contradictory to me. Can you clarify the wording here?
5. The decoupling between behavior and proximal policies is also used in the recently introduced MDPO [1] algorithm (Section 5.1). Another useful reference could be [2]. Particularly, if one is familiar with the NPG or Mirror Descent connection of PPO/PPG, it should be fairly straightforward to see that coupling the behavior and proximal policies is not a hard requirement (something that maybe is not as clear from the PPO paper).

References:

[1] Mirror Descent Policy Optimization.

[2] A functional mirror ascent view of policy gradient methods with function approximation.

**Summary Of The Paper:**

The paper proposes decoupling the behavior and proximal policies used in policy optimization algorithms such as PPO and PPG. Typically, the behavior policy itself is used as the proximal policy, i.e. the policy to which updates need to be close to (maintain trust region). In such a case, using stale or old data can lead to bad performance, leading to on-policy methods using data from the only most recent sampling iteration. The paper then talks about the batch invariance property, which refers to the performance remaining as is when changing the batch size. This is typically achieved by adjusting the optimization parameters such as learing rate (ex. if batch size is doubled, learning rate should be doubled as well). Finally, the authors perform experiments to test how the decoupling affects the usage of stale data, how batch invariance can be achieved with the decoupled objective, and how much better does the decoupling perform as compared to when the same policy is used as the behvior and proximal policies.

**Summary Of The Review:**

The paper does a good job in explaining the proposed method, in running appropriate experiments, and in terms of writing. However, it fails to motivate a single objective or problem which is addressed by the proposed method. I do not see what the main contribution is and so am currently recommending a weak reject. I am happy to revise the review if the authors can address these concerns.

---

> ### Author Response · Authors · 2021-11-17
> **Response to Reviewer kR2g**
>
> We are tremendously grateful to the reviewer for their generous and helpful review.
>
> One of the reviewer's key concerns was that the decoupled objective was not shown to significantly help achieve batch size-invariance. We agree that this was an important weakness, and have made a number of improvements to the paper to address this.
>
> The most important change is the introduction of a metric that allows us to measure the degree of batch size-invariance. Specifically, we look at the difference between the final normalized return at different batch sizes. This metric allows us to provide a more rigorous analysis of our ablations, and we show that the effects of all of the ablations are statistically significant at the 0.1% level.
>
> In particular, the difference between Figure 2 (left) and Figure 2(d) (bottom right) is statistically significant. However, this plot does not convey the difference well, since it averages over the environments, and there is an outlier environment (Heist). We have therefore added a new plot (Figure 3) of our metric, broken down by environment. This makes the effects of the ablations much clearer.
>
> We agree that our paper has two clear contributions: the decoupled objective, and batch size-invariance. However, we believe that these contributions are intertwined, and that our new analysis provides good evidence for this.
>
> In response to the minor comments and questions:
>
> 1. Fixed.
> 2. "Does this small enough batch size have anything to do with the critical batch size? I assume not right?" - It actually does. The reason that SGD with a batch size larger than the critical batch size does not achieve batch size-invariance is essentially that it is integrating the SDE using steps that are too coarse. We have added a comment in the paper reflecting this.
> 3. "Are you trying to say that more the policy differs a lot from the behavior policy, the more performance improvement we'll see?" - Not quite. The point is that for anything we do to control how fast the policy changes - such as a KL penalty or clipping - the behavior policy is irrelevant to this. "Trust region" methods try to keep the policy close to the behaviour policy specifically. On the other hand, natural policy gradient methods also control how fast the policy changes, but do not use the behavior policy to achieve this. Thus it seems that PPO is better described as a natural policy gradient method than a trust region method. We have tried to improve the explanation here.
> 4. "The first sentence talks about large batch sizes but the next line mentions advice regarding small batch sizes. They seem contradictory to me." - Fixed. The point was that one may be forced to use a small batch size in some new domain, and to figure out hyperparameters using batch size-invariance rules, one would first need to know which values work well at larger batch sizes.
> 5. "The decoupling between behavior and proximal policies is also used in the recently introduced MDPO algorithm (Section 5.1)." - Thank you for this reference. This is very relevant work and I have added a citation to it in the paper.

---

> > ### Comment · Reviewer_kR2g · 2021-11-27
> > **Thank you for the response!**
> >
> > The new Figure is certainely more interesting than Figure 2, with regards to the batch size invariance property. I am quite convinced that EWMA and the decoupling is indeed helping here. It might be good to note that ProcGen scores are quite close for different algorithms, and so the effect of the batch-invariant property might not be as apparent for these domains.
> >
> > On point 3 of minor comments: I think I agree. As long as the advantage estimates are with respect to the behavior policy, looks like there is a good amount of decoupling that can be achieved so as to remain batch-invariant. However, there must be a limit to this decoupling, since if the KL is encouraging to stay close to a policy very different from the behavior one, then the two terms can be at a conflict. I somehow feel this was not explicitly stated as the main contribution, but this might just be a subjective opinion.
> >
> > I am happy to advocate for an acceptance of this paper. As mentioned, the main story is batch-invariance and decoupling is something that allows us to achieve it with the current algorithms. I have revised my rating accordingly. Based on my calibration of reviews, I would ideally want to give a rating of 7 for this paper. However, since there is no '7' option, I am giving the benefit to the authors, with an 8 rating.

---

### Official Review · Reviewer_VVcj · 2021-11-02

**Correctness:** 3
**Technical Novelty And Significance:** 2
**Empirical Novelty And Significance:** 2
**Recommendation:** 1
**Confidence:** 5

**Main Review:**

Weaknesses:

1. In experiment 5.1, the author created artificial staleness data to evaluate their method. By comparing Figure 1(b) and Figure 1(c), we see that the vanilla PPO algorithm performs better than their proposed algorithm! If PPO with $\pi_{old}=\pi_{behavior}$ is much better than the proposed algorithm, why does the author want us to use their method?

2. By comparing Figure 2(a) and Figure 2(d),  we see that the most critical factor is the Adam step size instead of the proposed EWMA method. I have no idea the meaning of the proposed EWMA method, and we can obtain nearly nothing when using the EWMA method. And the way of how to adjust the learning rate of Adam is not new. So, what is the contribution of this paper?

3. In Figure 3,  we see EWMA has minimal improvement on current deep RL methods. I doubt the value of this work.



**Summary Of The Paper:**

This paper proposed a new method to deal with batch size-invariance for policy optimization.

**Summary Of The Review:**

This paper wants to propose a new deep RL method, which makes use of their decoupled objectives. However, I can see slight improvement over their methods. And the proposed EWMA algorithm contributes little to the batch size invariance.

---

> ### Author Response · Authors · 2021-11-17
> **Response to Reviewer VVcj**
>
> We are very grateful to the reviewer for their candid review.
>
> We have now released an updated version of the paper which we hope addresses some of their concerns. The biggest change is the introduction of a metric that allows us to measure the degree of batch size-invariance. This allows us to provide a more rigorous analysis of our ablations, and we show that the effects of all of them are statistically significant.
>
> In response to the list of weaknesses:
>
> 1. With vanilla PPO, the introduction of even a tiny amount of staleness (1 iteration) impacts performance, even from the very start of training. In contrast, with the decoupled objective, staleness can be increased much further (to 8 iterations), with very little impact on performance until the very end of training. Vanilla PPO's lack of robustness to small amounts of staleness is a significant weakness that we have managed to fix, with obvious practical benefits.
> 2. It is true that the effect size of our final ablation is smaller than the other ablations, which reflects the fact that PPG is more robust to changes in the KL penalty than to changes in other hyperparameters such as the step size. However, our new analysis shows that the effects of all of our ablations have statistically significant effects. We have also added a new plot breaking down the effect of each ablation on the individual environments (Figure 3), which makes the effects of the ablations much clearer.
> 3. It is true that the improved performance of the EWMA is small, but it is remarkably consistent, outperforming the baseline in 14 of 16 environments for PPO and 15 of 16 environments for PPG. Importantly, the additional hyperparameter β_prox was tuned only on the first 8 environments. We have added an additional plot and further explanation to make both of these points clearer. We deliberately did not make this improved performance the main focus of the paper, because of the small effect, but think that the result is interesting enough to be worth discussing.

---

### Official Review · Reviewer_tiiD · 2021-11-05

**Correctness:** 2
**Technical Novelty And Significance:** 3
**Empirical Novelty And Significance:** 3
**Recommendation:** 5
**Confidence:** 4

**Main Review:**

This paper takes on an essential issue in reinforcement learning in designing algorithms that require less hyperparameter tuning. The ideas in this paper could be beneficial to the community and make it easier for practitioners to use these algorithms. This paper could be excellent, but as currently written needs some significant improvements. The main two areas where the paper can be improved are the clarity of presentation and the design of the experiments.

Clarity:
As it is currently presented, many of the methods and ideas are described with vague or verbose language, making it more challenging to interpret the paper's main ideas. I will detail some of these below, but the central theme of these changes is that the paper needs to be more precise in its statements. These improvements can largely be achieved by introducing mathematical definitions and providing formal statements of the methods.

"we can preserve behavior, as a function of the number of examples processed, by changing other hyperparameters (as long as the batch size is not too large)" -- what does the batch size being too large mean? The batch size limit should be quantified so that a user can understand what large enough actually means.

Definition of L^{KLPEN} uses the symbol \hat E_t. What is the \hat E? This is not a common mathematical symbol and is never defined.

The critical step size is mentioned often in the paper, and a description of it is given in section 3.1, but a formal definition would remove any ambiguity of the interpretation.

In Section 3.1, the derivational derivative symbol \nabla_\theta is used for what is meant to be a partial derivative. Note that the gradient is the vector of partial derivatives with respect to all inputs to a function, not just a single one. Following standard mathematical symbols would make this clearer.

"If the batch size is small compared to the critical batch size, then the difference between θ t and θ t+1 is mostly noise, and moreover this noise is small compared to the total noise accumulated by θ_t over previous updates." -- what does mostly noise mean? Do second-order effects not impact the process? For a paper studying batch size invariance, it is light on providing precise details of the source of errors in these approximations.

"To compensate for the batch size being divided by c," -- "c" is never defined in the main paper or the appendix. The rest of the statements cannot be correctly interpreted.

"Hence Adam is effectively dividing the learning rate by √ c automatically, and so the step size α only needs to be adjusted by an additional √ c to effectively divide the learning rate by c overall." -- This is the perfect example where having mathematical definitions of the operations or pseudo code of the invariant update would drastically improve clarity.

"PPO is automatically optimization batch size-invariant, as long as the optimization algorithm it uses (such as SGD or Adam) is as well." -- This is unproven and contradicts statements that the proximal policy is needed for batch size invariance.


"This demonstrates that the decoupling the proximal policy from the behavior policy is safe and useful." -- The use of the word "safe" here is probably best avoided since it has many meanings that can be misinterpreted.



Experiments:
The primary issue with the experiments is that they are not designed to show batch size invariance directly. Instead, they focus on seeing if the algorithm has similar performance for different batch size parameters. While having a similar performance for different batch sizes is an expected property of the methods, these experiments cannot determine when batch size invariance should be expected or when it fails. This shortcoming of the experiment design is revealed on page 6 when the paper provides a guess at the failures and offers no empirical evidence. The results do indicate that there is some batch size invariance present. The paper should make it clear what the limits of batch size invariance are. Other comments on the experiments are provided below.

The experiments use the Procgen environments, which are computationally expensive to use due to image-based features. The result is that only a few trials of each algorithm are run on each environment. Since it is widely known that the performance of RL algorithms has high variance, there are not enough trials being executed to make any claim of significance. Furthermore, since the contributions of this paper have nothing to do with performing well in such complex environments, the paper could greatly benefit from using smaller, more controllable environments and averaging out the effects of noise (Ceron and Castro, 2021).

Figure 2b) "The decoupled objective allows the correct importance sampling ratio to be used while maintaining the age of the proximal policy, preventing performance from degrading much until the data is very stale." -- This is not a result but a hypothesis. No experiment shows that the age of the proximal policy is maintained. There could be many other factors that lead to performance not being degraded.

"We were able to achieve complete batch size-invariance (with essentially indistinguishable learning curves at every batch size)" -- This statement is not proven and imprecise. While the learning curves are close together, there is no uncertainty quantification, so these results cannot be trusted. To make such a claim, one needs to have a proper definition of batch size invariance, the corresponding metric, a definition of what complete means, and proper uncertainty quantification.

"One possible reason is that we did not adjust the Adam β1 and β2 hyperparameters, although we did not ﬁnd the adjustment proposed in Section 3 to noticeably help. We leave further analysis of this to future work." -- Were experiments run that show this, or is this a guess? If there is an experiment, then it should be shown. This is also a fundamental experiment that should beconducted to show what is necessary for batch size invariance. Skipping it shows that this work is incomplete.

"Next most important is the advantage normalization adjustment, which does not matter much in many environments, but matters a lot at the smallest batch sizes in environments for which the advantage standard deviation estimates are particularly noisy." -- Was advantage normalization discussed anywhere in the paper? I could not find it.

There are ablations trying to show what components of the proposed algorithms are necessary for batch size invariance. However, it is unclear how it is being determined if a method has a significant impact on batch size invariance. The effect and uncertainty of each choice needs to be quantified. If the EWMA is unnecessary, how is any discussion about the "proximal" policy-relevant to batch size invariance?

The performance comparison of the algorithms is irrelevant and completely dependent on the choice of hyperparameters. If one wants to compare the performance of these algorithms, then this should be taken into account. The complete algorithm definition idea proposed by Jordan et al. (2020) seems like a good fit for conducting the experiments in this type of work.

In section 6.2, much "advice" is given on these algorithms, but it is unclear if any experiments support this advice. These heuristics need proper evaluation before they should be recommended in a formal academic context.


Ceron, Johan Samir Obando, and Pablo Samuel Castro. "Revisiting Rainbow: Promoting more insightful and inclusive deep reinforcement learning research." International Conference on Machine Learning. PMLR, 2021.

Jordan, Scott, et al. "Evaluating the performance of reinforcement learning algorithms." International Conference on Machine Learning. PMLR, 2020.


**Summary Of The Paper:**

Standard reinforcement learning algorithms such as PPO have several "batch size" hyperparameters parameters, and changing them impacts how one needs to choose the step size. To better understand these algorithms and their hyperparameters, this paper investigates batch size invariance, a property that allows the algorithm's behavior to remain constant for changes to the batch size through the modification of other hyperparameters. Batch size invariance has been studied significantly in supervised learning, but these PPO-style algorithms have updates that break the previous batch size invariance techniques. This paper develops two new algorithm variants that provide a way to achieve batch size invariance. Empirical results show that the methods are somewhat effective at providing batch size invariance.


**Summary Of The Review:**

Due to the lack of precision and weaknesses in the experiments, I cannot recommend this paper for acceptance, but I do acknowledge that the ideas may be helpful.

---

> ### Author Response · Authors · 2021-11-17
> **Response to Reviewer tiiD**
>
> We are extremely grateful to the reviewer for their thorough and thoughtful review. We have made many improvements to the paper in response to their comments.
>
> The biggest change to the paper is the introduction of a metric that allows us to measure the degree of batch size-invariance. Specifically, we look at the difference between the final normalized return at different batch sizes. This difference being small is a necessary condition of a high degree of batch size-invariance, although it is of course not a sufficient condition, as the reviewer rightly points out.
>
> This metric allows us to provide a more rigorous analysis of our ablations. Using a van Elteren test (a stratified version of the Mann-Whitney U test, see LaVange & Koch (2006)), we test the null hypothesis that the ablation had no effect on the difference in final return between the smallest and largest batch sizes in any of the environments. For all of the ablations, we rejected the null hypothesis at the 0.1% level (including a Bonferroni correction to account for multiple comparisons, which is conservative).
>
> Although the reviewer hypothesized that we did not use enough trials to make any claim of significance, the large number of environments used gives us enough statistical power to make a strong claim of significance.
>
> The additional plot of the new metric broken down by environment (Figure 3) is also informative.
>
> Another important change we have made is to include experiments on the β_1 and β_2 hyperparameters, which are now given in Appendix E.
>
> In addition to these changes, we have tightened up many imprecise claims. Most notably, we have removed all mentions of "complete" batch size-invariance, which was not defined. Instead we now refer to the degree of batch size-invariance, which is measured by our new metric.
>
> In response to some of the other comments:
>
> - "What is the \hat E?" - We have now included the definition of this (it indicates the empirical average over a finite batch of timesteps).
> - "The derivational derivative symbol \nabla_\theta is used for what is meant to be a partial derivative." - The directional derivative is correct - this is a vector equation. We believe that gradient descent equations are very commonly written in vector form.
> - "What does mostly noise mean?" - The claims in the paper about batch size-invariance for SGD are well-established in the literature, and the explanation we provide is only intended as a sketch for the benefit of the reader's intuition. However, we did not make this clear enough. We have now clearly signposted this explanation as a sketch and referred the reader to Mandt et. al. (2017) for a more thorough explanation.
> - "'c' is never defined in the main paper or the appendix" - "c" is a dummy variable. We have now included the phrase "some constant" to indicate this more clearly.
> - "Hence Adam is effectively dividing the learning rate by √ c automatically" - This explanation was crammed and confusing, so we have expanded it and moved it to Appendix D to avoid distracting from the main focus of the paper.
> - "This is unproven and contradicts statements that the proximal policy is needed for batch size invariance." - The qualifier "optimization" in the phrase "optimization batch size" is crucial here, changing the claim from a contradiction to something that is true by definition. We have updated the wording here to try to make this clearer.
> - "The use of the word 'safe' here is probably best avoided" - Agreed, this has now been fixed throughout.
> - "Was advantage normalization discussed anywhere in the paper?" - This is mentioned in Section 4. We have expanded the discussion of it.
> - "The performance comparison […] is completely dependent on the choice of hyperparameters" - This is an excellent point that we should have addressed. In fact, the additional hyperparameter β_prox was tuned only on the first 8 of the 16 Procgen environments, and so the algorithms are "complete" on the last 8 environments in the terminology of Jordan et. al.
>
> Lisa M LaVange and Gary G Koch. "Rank score tests." Circulation, 114(23):2528–2533, 2006.
>
> Stephan Mandt, Matthew D Hoffman, and David M Blei. "Stochastic gradient descent as approximate Bayesian inference." arXiv preprint arXiv:1704.04289, 2017.

---

### Author Response · Authors · 2021-11-17
**Summary of changes**

We are incredibly grateful to all of the reviewers for their detailed and insightful comments. We have released a revised version of the paper with many improvements thanks to these. A summary of the main changes is as follows:

- The biggest change is the introduction of a metric that allows us to measure the degree of batch size-invariance. Specifically, we look at the difference between the final normalized return at different batch sizes. This metric allows us to provide a more rigorous analysis of our ablations, and we show that the effects of all of the ablations are statistically significant at the 0.1% level. We also include a plot of this metric broken down by environment (Figure 3), which better conveys our results than the existing plot of mean normalized return.
- We have included additional experiments on the β_1 and β_2 hyperparameters, which are given in Appendix E.
- We have tightened up many imprecise claims. We have removed all mentions of "complete" batch size-invariance, which was not defined, and instead we now refer to the degree of batch size-invariance, which is measured by our new metric. We have also more clearly marked the sketch explanation in Section 3.1 as such, and referred the reader to Mandt et. al. (2017) for a more thorough explanation.
- We have explained how we tuned the β_prox hyperparameter, which makes our EWMA comparison results more meaningful.
- We have given additional motivation for the use of an EWMA.

Stephan Mandt, Matthew D Hoffman, and David M Blei. "Stochastic gradient descent as approximate Bayesian inference." arXiv preprint arXiv:1704.04289, 2017.

---

### Decision · Program_Chairs · 2022-01-20

**Decision:**

Reject

**Comment:**

This paper studies the method to achieve the batch size-invariant for policy gradient algorithms (PPO, PPG). The paper achieves this by decoupling the proximal policy from the behavior policy. Empirical results show that the methods are somewhat effective at providing batch size invariance.

After reading the authors' feedback, the reviewer discussed the paper and they did not reach a consensus. On the one hand, the rebuttal made some reviewers change their minds who appreciated the explanations provided by the authors and the new Figure that better highlights the batch size invariance property.
On the other hand, some reviewers think that there is still significant work to be done to get this paper ready for publication. In particular, it is necessary to improve the theoretical analysis and the evaluation of the empirical results.

I encourage the authors to follow the reviewers' suggestions while they will update their paper for a new submission.